# Gold Nanoparticles as a Potent Radiosensitizer: A Transdisciplinary Approach from Physics to Patient

**DOI:** 10.3390/cancers12082021

**Published:** 2020-07-23

**Authors:** Sébastien Penninckx, Anne-Catherine Heuskin, Carine Michiels, Stéphane Lucas

**Affiliations:** 1Research Center for the Physics of Matter and Radiation (PMR-LARN), Namur Research Institute For Life Sciences (NARILIS), University of Namur, Rue de Bruxelles 61, B-5000 Namur, Belgium; sebastien.penninckx@unamur.be (S.P.); anne-catherine.heuskin@unamur.be (A.-C.H.); stephane.lucas@unamur.be (S.L.); 2Unité de Recherche en Biologie Cellulaire (URBC), Namur Research Institute For Life Sciences (NARILIS), University of Namur, Rue de Bruxelles 61, B-5000 Namur, Belgium

**Keywords:** gold nanoparticle, radiosensitizer, protontherapy, mechanism, oxidative stress, clinical translation, nanomedicine

## Abstract

Over the last decade, a growing interest in the improvement of radiation therapies has led to the development of gold-based nanomaterials as radiosensitizer. Although the radiosensitization effect was initially attributed to a dose enhancement mechanism, an increasing number of studies challenge this mechanistic hypothesis and evidence the importance of chemical and biological contributions. Despite extensive experimental validation, the debate regarding the mechanism(s) of gold nanoparticle radiosensitization is limiting its clinical translation. This article reviews the current state of knowledge by addressing how gold nanoparticles exert their radiosensitizing effects from a transdisciplinary perspective. We also discuss the current and future challenges to go towards a successful clinical translation of this promising therapeutic approach.

## 1. Background 

Radiotherapy (RT) has emerged in the past century as one of the most efficient cancer treatment modalities. Nowadays, over 50% of patients receive RT for curative or palliative purposes over the course of their treatment [1,2]. This technique consists of delivering lethal doses of ionizing radiation (IR) into the tumor via an external beam (conventional RT) or from an internally implanted radiation source (brachytherapy). The cell exposure to IR generates a succession of processes that differ in time-scale (Appendix A). First, the physical phase occurs until 10^−12^ s after irradiation. It describes the energy deposition and its spatial pattern following interactions between IR and the atoms composing cells. IR interacts mainly with orbital electrons, ejecting some of them from atoms (ionization) and raising others to higher energy levels within the atom (excitation). If ejected electrons are sufficiently energetic, they excite or ionize surrounding other atoms, leading to a cascade of ionization events. Energy deposition and fast relaxation processes that occur in this phase lead to the formation of various ionized or excited water molecules. These molecular species, called free radicals, are highly instable chemicals due to the presence of an unpaired electron in their outermost valence shell. Water radicals, usually called reactive oxygen species (ROS), can be stabilized via oxidation-reduction reactions with other molecules in the surrounding medium. This stage that describes the generation, diffusion, and reaction of free radicals, is usually called the chemical phase. It occurs at longer time scales than the physical phase (until about 10^−6^ s). Amongst these reactions, ROS react with different biomolecules such as DNA, lipids, and proteins. DNA is the most critical target because there exist only two copies of each of the DNA molecules. ROS reaction with DNA produce DNA damages through the so-called “indirect effect”. It has to be noted that ionization-excitation processes that occur in the physical phase can lead to direct DNA ionization and hence to DNA damage production through the “direct effect”. Finally, the biological phase includes all subsequent processes that are too various to be listed here. Amongst these biological responses, DNA damage repair pathways start in short time-scale after exposition (seconds). According to Curtis model [3], cells containing lethal lesions will die while cells containing potentially lethal damages (PLD) will try to repair these DNA damages. This can lead to the cell survival after a correct DNA repair or to the cell death if a misrepair on key survival genes occurs. All the cellular DNA repair processes end after a few days. It has to be noted that the survival or death of cells can occur later because cells may undergo multiple mitotic divisions before dying. 

In radiobiology, five factors (described as “5 R’s of radiotherapy”) have been identified as capable of modifying the cell responses to radiation, influencing the cancer cell death and so the success of radiotherapy [4,5]. These factors and their time-dependency are the basis for modern fractionation radiotherapy:Repair: contrary to proteins and lipids, which can also be damaged by IR, genomic DNA molecules are present in the cell nucleus in only two copies, making them a critical target. To respond to a plethora of physical, chemical, or biochemical genotoxic agents that are able to damage DNA, cells have complex mechanisms enabling DNA repair. The DNA repair systems keep the level of DNA damages as low as possible reducing cell death and mutation induction probabilities. In the radiotherapy context, this means a reduction in treatment efficiency. Thereby, various chemical agents (etoposide, gemcitabine, etc.) were designed to interfere with DNA repair processes in order to enhance the effect of IR [6].Redistribution: cell radiosensitivity considerably varies with the phases of the cell cycle. Cells in the S phase are the most resistant while cells in late G_2_ and M phases are the most sensitive [7]. The reason for the resistance in S phase is thought to be a higher homologous recombination capacity due to a greater availability of the undamaged sister chromatid, used as a template. Moreover, conformational changes in DNA during replication facilitate an easier access for the repair complexes [5]. In contrast, the greater sensitivity in late G_2_ and M phases is due to the start of mitosis with DNA damages, leading to quicker cell death. Thereby, the fractionation therapy, which takes place over multiple sessions, is more effective since it enables the cells in the G_1_ and S phases of the cell cycle to move towards more radiosensitive phases.Repopulation: repopulation of cancer cells has been considered as the main cause of radiotherapy failure [5]. Various chemical agents able to inhibit or slow down cellular growth and proliferation are used to counteract this effect [6].Reoxygenation: oxygen plays a key role in radiation therapy through the ROS production during the chemical step as well as for the DNA free radicals fixation [5]. During the uncontrolled proliferation of cancer cells, the tumor quickly exhausts the oxygen supply from the normal vasculature resulting in the generation of hypoxic areas. Cox et al. [8] estimated that the proportion of hypoxic cells in a tumor usually ranges from 10% to 15%. When a tumor is irradiated, the oxygenated cells are killed more easily than hypoxic ones due to the higher oxygen pressure within them. After irradiation, the proportion of hypoxic cells in the tumor is thus higher than prior to irradiation. Nevertheless, the situation is not static: irradiation also triggers the nitric oxide synthase activation enabling an arterial vasodilatation which increases the tumor tissue perfusion [9]. This phenomenon allows tissue reoxygenation during a given time period after the irradiation. If the interval between radiation doses is long enough to allow reoxygenation to take place, then originally hypoxic cells become oxygenated again and are more radiosensitive to the next dose. Thereby, the presence of hypoxic cells has a lower effect on the treatment success if the total dose is divided into fractions enabling a sufficient duration for cell reoxygenation in between the fractions [5].Intrinsic Radiosensitivity: it has been evidenced that patient-related factors account for as much as 80–90% of the variation observed in patient response to radiation [10]. Nowadays, the origin of this variability remains poorly understood but it is generally accepted that it is due to genetic variations.

Although key parameters driving the success or failure of radiotherapy have been identified, modern RT is still limited by the side effects caused to healthy tissues surrounding a solid tumor. One of the current challenges is to maximize the differential response between tumor and healthy tissues. The likelihood for a tumor to be controlled is called the tumor control probability (TCP), while the one for the healthy tissue side-effect is called normal tissue complication probability (NTCP) [11]. These probabilities are represented by a sigmoid function according to the dose, as illustrated in Figure 1. From these definitions, it follows that the probability of cure without complication (PCWC) is given by Equation (1): (1)$$PCWC=TCP\;\left(1-NTCP\right)$$

In Figure 1, it appears that dose associated with tumor eradication is not very different from the dose associated with normal tissue complication development. Therefore, improvements have been made over the past few decades with the implementation of advanced precision technologies such as intensity modulated radiotherapy (IMRT) or the use of charged particles instead of classical X-rays. Although these advances improve the dose conformity and the clinical outcome, additional therapeutic benefit may be gained by using RT combined with chemotherapy. Uncommon in the early 2000s, it already has reached 10% of all RT treatments in 2013 [13]. In this context, the use of a molecule localized into the tumor, which has the ability to increase cancer cell killing, can move the TCP curve towards the left. Consequently, a significant increase in the maximum and width of the PCWC distribution generates a larger margin for the therapeutic window (Figure 1). These kinds of molecules are called “radiosensitizers” and enable to reach a given tumor cell killing using a reduced total dose delivered to the patient. 

Numerous studies have shown that high atomic number (Z) nanomaterials can play the role of radiosensitizer. Nowadays, the most studied material is gold nanoparticles (GNPs) due to their high X-ray absorption coefficient, good biocompatibility, and ability to improve the performance of magnetic resonance imaging diagnosis [14]. In the pioneering work of Hainfeld et al., injections of 1.9-nm GNPs showed an increase in the survival of mice bearing EMT-6 mammary carcinoma in combination with 250 kVp X-rays compared to X-rays alone [15]. Since this work, many in vitro and in vivo studies have evidenced the ability of high-Z nanoparticles, injected into the tumor, to amplify RT efficiency. In this review, we summarize the main in vitro works regarding the ability of spherical GNP to sensitize cells to kilovoltage (kV) and megavoltage (MV) X-rays as well as to protons. This works aims at shedding light on mechanisms responsible for the GNP radiosensitization effect and at addressing the current and future challenges towards a successful clinical translation. 

## 2. Evidence for GNP Radiosensitization Effect

The main in vitro works regarding GNP radiosensitization effects in literature are summarized in Table 1, and reveal immediately the use of different GNP size, GNP surface functionalization, cell model, and radiation quality. Furthermore, the high-Z nanoparticle-mediated enhanced radiation sensitivity has been reported by several groups with different indicators. 

In order to provide a comprehensive understanding of the response to radiation resulting from the presence of GNPs, it is therefore mandatory to adopt a common language. As recommended [5], we express the GNP effect on radiation response in terms of a “sensitization enhancement ratio” (SER) defined as: (2)$$SER_{x}=\frac{Radiation\;dose\;without\;GNPs\;to\;achieve\;x}{Radiation\;dose\;with\;GNPs\;to\;achieve\;x}$$

The SER is calculated for a given biological effect (a 10% survival fraction in this work) and has the advantage to take into account the entire dose-response curve. Values above 1 mean that the drugor nano-object of interest enhances the cell death in comparison to the treatment without it. 

In Table 1, we only reported studies that present data regarding the GNP content in cells and an entire survival curve, e.g., an IR dose-response analysis. This last criterion enables a description of cell survival using the linear-quadratic model defined as:(3)$$SF=\exp\left(-\alpha D-\beta D^{2}\right)$$

This model describes the survival fraction (SF) as a function of the dose (D) using two parameters α (Gy^−1^) and β (Gy^−2^). Curtis suggested a radiobiology significance of these parameters in his unified repair model [3]. The non-repairable lesions produce single-hit lethal effects resulting in the linear component of survival fraction (exp (−αD)). The repairable lesions depend on the competition between the repair and the binary misrepair processes leading to a quadratic component in cell survival (exp (−βD^2^)). Therefore, α/β ratio is frequently used to predict clinical effect of irradiation since it corresponds to the dose at which the linear component (related to non-repairable lesions) equals the quadratic contribution (related to PLD).

Chithrani et al. [20] reported a radiosensitization effect on HeLa cells using a combination of GNPs and 220 kVp X-rays. They investigated the impact of GNP size on the enhancement effect and demonstrated that a greater SER was obtained with 50 nm than with 14 and 74 nm GNPs. The authors explained this observation by a higher cell uptake of 50 nm GNPs compared to smaller and bigger GNPs. The same group evidenced that the SER decreases from 1.57 to 1.17 by increasing the energy beam from 105 kVp to 6 MV X-rays, respectively. Enferadi et al. [16] reached the same conclusion by studying ALTS1C1 cells containing 1.8 nm GNPs exposed to various X-rays energy. The importance of surface functionalization was shown by Ma et al. [21] who exposed GNPs coated with polyethylene glycol (PEG), folic acid or TAT peptide to 6 MV X-rays. Although significant radiosensitization effects were reported for all GNPs used in the study, the ones coated with TAT peptide exhibited the highest SER (2.3 compared to 1.5 for the PEG-coated ones). The authors explained this observation through a positive correlation between SER and cellular uptake of GNPs. Several groups have focused their GNP-sensitization studies on commercial 1.9 nm GNPs (Aurovist^TM^, Nanoprobes, Yaphank, NY, USA). Following 225 kVp X-ray exposure, Butterworth et al. [32] demonstrated the ability of these GNPs to radiosensitize a large variety of cells from T98G glioblastoma cells (SER = 1.85) to MDA-MB-231 breast cancer cells (SER = 1.22). It was reported that some cell lines, such as human prostate DU145, are not sensitized by these 1.9 nm GNPs even if significant gold uptake was observed (SER = 1.03 [32] or 1.08 [33]). However, the absence of gold content quantification in this study prevents going forward in the process understanding. The differences in SER reported in Table 1 cannot be streamlined due to the diversity of parameters and conditions tested in literature as well as to a lack of data regarding key parameters. Indeed, the cellular gold content upon irradiation was pointed as a key parameter of GNP radiosensitization effect but this information is missing in the majority of works leading to a relatively low number of studies described in Table 1.

## 3. Mechanisms Associated to GNP Radiosensitization Effect

### 3.1. Physical Enhancement

Originally, the rationale for using high-Z materials as radiosensitizers was based on their ability to increase the dose deposited in the target volume due to differences in their mass energy absorption coefficient by comparison with soft tissue. If the reader is not familiar with the main concepts of radiation–matter interaction, one can refer to radiation physics textbooks [34,35]. 

In water, the most probable mechanism by which clinical X-rays lose their energy is the Compton effect, which leads to photon scattering and to the ejection of electrons from the atom. The scattered photon, which has a reduced energy compared to the incident one, can interact through the photoelectric effect where it is wholly absorbed by a bound electron leading to its ejection from the atom. When the photon interacts with high-Z materials, such as gold (Z = 79), the total absorption cross-section is larger due to the higher number of electrons per atom (Figure 2A). Both photoelectric and Compton effects result from interactions with electrons, so, higher Z materials correspond to higher numbers of electrons per atom. Therefore, the increased attenuated cross-section observed in Figure 2A evidences that high-Z materials absorb substantially more energy per unit mass than water does when X-rays pass through them. By dividing the gold absorption cross-section by the one of water, we observe that gold can be 100 times more effective at absorbing photon energy than water. This effect is translated into a significant higher local dose proportional to the gold weight percent in the medium (Figure 2B). 

If we consider charged particles instead of photons, the radiation–matter interaction changes. When a charged particle passes through water, it interacts by collisions with electrons and nuclei of oxygen and hydrogen atoms constituting water. It has to be noted that the nuclear contribution is negligible for protons at energies used in clinics (<230 MeV) meaning that the energy loss is mainly driven by inelastic Coulomb collisions with the electrons (electronic contribution). Figure 2C shows the proton total stopping power in water and in gold depending on the proton energy. The gold stopping power of protons is larger than the one of water. The electronic stopping power has only a weak dependence on Z through the logarithm of mean ionization potential (see Bethe–Bloch expression) [35]. By comparing the stopping power for the two media, we observe that the stopping power ratio is proportional to the proton energy and is up to 10-fold higher for gold compared to water for a 200 MeV proton beam (Figure 2D). This reflects the difference of one order of magnitude between mean ionization potentials of gold (≈790 eV) and water (≈75 eV). This means that GNP physical enhancement is mainly due to the density difference between gold and tissue in the case of charged particles while it is related to a strong Z-dependence of photoelectric effect in the case of kV photons. 

To quantify the physical enhancement, the dose enhancement value (DE) can be used [38]:(4)$$D_{GNP}=D_{water}+\left(\;D_{water\;}*\;DE\;\right)\;DE=\frac{D_{GNP}}{D_{water}}-1$$

This dose enhancement value is expressed in arbitrary units called dose enhancement units (DEU). An enhancement of 1 DEU means that the delivered dose doubles when GNPs are present in the medium. As shown in Figure 2E, the DE depends on the GNP amount in the cell leading to a maximal theoretical enhancement of 1 DEU per gold weight percent (=1 DEU.WP^−1^) for kV X-rays and 0.09 DEU per gold weight percent (=0.09 DEU.WP^−1^) for protons.

Due to the aforementioned processes, energy deposition is higher in gold than in water. This higher local dose deposition triggers low-energy electron (LEE) emission from GNPs, which subsequently deposit their energy in the surrounding medium leading to potential increase in DNA damage and subsequently in cell death. This suggested cascade of events highlights that high-Z nanoparticles can play the role of radioenhancers through a physical enhancement mechanism. However, physical enhancement is difficult to verify experimentally due to technical issues. Indeed, the most straightforward way to measure this should be a measurement of LEE emission from GNPs in the medium of interest. Although possible, these experiments involve the use of complex indirect measurement using chemical or biological reactions. Casta et al. [39] measured the electron emission from 6 nm GNPs and gold plane surface in vacuum after X-ray irradiation. The authors demonstrated that the electron emission from GNPs is 2.3-fold more important than in the case of gold surface and display a prominent peak below 100 eV. Since it has been demonstrated that low-energy electrons could be very efficient at causing damage to DNA molecules through dissociative electron attachment [40], these findings are in agreement with the suggested mechanism hypothesis. Another experimental evidence of this mechanism is the work of Brun et al. [41] who studied the influence of GNPs on DNA damage generation under irradiation. Supercoiled plasmid DNA and GNPs were mixed together prior to be exposed to kV X-rays. Subsequently, a gel electrophoresis was performed to determine the DNA damage. Indeed, when damages occur in supercoiled plasmid DNA, its spatial conformation changes leading to a difference in its ability to migrate in a polymeric gel. The authors reported an enhancement of 0.48 DEU for an equivalent gold in water of 0.4 WP, giving rise to a 1.2 DEU.WP^−1^. 

In addition to these indirect physical measurements, theoretical simulation programs can calculate LEE emission and the subsequent physical enhancement. Early models investigated the macroscopic dose enhancement induced by the presence of GNPs randomly distributed in a volume. Several authors evidenced an energy-dependent physical enhancement. Cho et al. [42] have computed a 1.5, 0.9, and 0.01 DEU.WP^−1^ when GNPs were exposed to 140 kVp, 250 kVp, and 6 MV X-ray, respectively. These results were confirmed by Mesbahi et al. [43] who also observed a strong energy dependency with enhancements from 1.4 DEU.WP^−1^ at 90 keV to 0.03 DEU.WP^−1^ at 660 keV. By studying the electron emission, Carter et al. [44] showed that the majority of electrons that escape a 3-nm GNP after X-rays exposition are LEEs (<100 eV). The number of these species is significantly reduced when the NP size increases. Moreover, they demonstrated that energy deposition decreases quickly as the distance increases, by nearly three orders of magnitude from 1 to 10 nm. Dose enhancement associated to long-range photoelectrons which can travel up to micrometers in water was negligible. Due to this nanoscale nature of dose enhancement, more recent studies considered microscopic dose distributions instead of classical macroscopic analyses to obtain information in the area close to the GNP surface. McMahon et al. [45] performed an energy deposition study for 40 keV X-ray irradiation by comparing the macroscopic and microscopic approaches. They computed a macroscopic enhancement of 0.075 DEU for a loading of 0.05 WP (1.5 DEU.WP^−1^) while they reported a 1.05 DEU (21 DEU.WP^−1^) at 2 nm away from the gold-filled region. This led them to suggest that several huge dose enhancement peaks located in close vicinity of the nano-objects cause the GNP radiosensitization effect. Furthermore, gold mass attenuation coefficient depends on the material thickness meaning that GNP size has an impact. Several groups evidenced that smaller nanoparticles deposit larger doses in their vicinity than bigger ones due to their greater surface to volume ratio [45]. Lin et al. [46] simulated the number of electrons emitted from a 50 nm GNP after X-ray exposition and reported very low values in the 10^−5^–10^−4^ range per incoming photon at best. 

Similar theoretical studies performed using protons showed a negligible macroscopic physical enhancement [47,48,49,50]. Martinez-Rovira et al. [50] performed simulations to evaluate the dose enhancement when protons pass through a medium containing GNPs. They did not report a significant energy deposition increase for a realistic configuration of the model. In another study, an energy-dependent emission of electrons from GNP surface was reported [48]. For a 5 nm GNP, they reported 8% and 20% increases in number of electrons ejected per incident proton after interaction with 1.3 and 4 MeV protons, respectively, compared to irradiation in the absence of GNPs. This energy dependency can be related to the higher number of projectiles needed to reach a given dose with higher proton energy. The authors highlight the relatively low energy (below 1.5 keV) of electrons and their highly trapped proportion (around 50%), meaning that half of these electrons cannot reach the NP surface. Moreover, the authors did not report a signature of GNPs in the macroscopic dose delivered. Similar results, obtained by Cho et al. [49], showed that the average dose enhancement over the entire solution volume was negligible even though an electron emission was reported. Finally, Sotiropoulos et al. [51] used a cell model where realistic GNP distribution was implemented to study the DNA damage under proton irradiation. They showed that, independently of the proton energy, the GNP size, the GNP concentration, and the GNP distribution, the physical enhancement is negligible. In conclusion, one can think that the low reported ability of GNPs to increase the macroscopic dose deposition of a charged particle prevents their potential use as radiosensitizer for charged particles. This is only true if the radiosensitization effects of GNPs are uniquely due to physical effects.

In order to investigate if this conclusion holds, we compared the predicted physical enhancement in in vitro studies (Table 1) to observed SER and maximal dose enhancement predictions. For that, these two parameters were plotted according to the gold weight percent (Figure 3). This figure highlights three main deviations from the dose enhancement physical predictions.

Several studies reported significant radiosensitization effects for cells containing less than 0.1 WP of GNPs, conditions that are associated to insignificant dose increase prediction for X-rays (<0.1 DEU). For example, Liu et al. [24] investigated the effect of 16 nm tirapazamine conjugated GNPs on HepG2 liver cancer cells under X-ray irradiation. The authors reported a gold content of 650 GNPs/cell corresponding to 0.001 WP and a significant enhancement (0.25 DEU) in these conditions. This observed radiosensitization effect is 250-fold higher than the predicted one (=0.001 DEU).According to the theory, no increase in overall dose deposition would be expected using MV X-rays. However, various studies have reported significant radiosensitization effects [19,22,27,52,53]. In the same way, it is interesting to note that radiosensitization effects were reported using proton beam while physical enhancement calculation predicted only negligible dose enhancement.Finally, the observed enhancement values are generally higher than the predicted ones as illustrated in Figure 3. For almost all experimental results, the in vitro observed enhancement is higher than the corresponding maximal physical predicted dose increase (plotted as dashed or dotted line). Moreover, the correlation between the predicted dose enhancement and observed radiosensitization is not significant (*p*-values are 0.69, 0.43, and 0.32 for kV X-rays, MV X-rays, and protons data, respectively).

In the light of these observations, the simulation tools have to be improved in order to progress in the understanding of this physical mechanism. The first observation that can be made is that predictions rely on incomplete physical events. Monte Carlo codes used to evaluate physical enhancement are often based on incomplete Auger cascades. Most of publications reports the use of Geant4s. However, since its deexcitation module is only available up to the M shell, some vacancies created in outer shells are not taken into account leading to an underestimation of dose enhancement. This is a major issue for simulations using charged particle where the large majority of ionizations will take place in the outer shells. Moreover, Geant4 models do not consider the emission of electrons below the mean ionization potential of the medium (790 eV for gold) leading to an underestimation of the LEE yield. This underestimation was recently evidenced by comparing experimental and simulation data about GNP exposed to protons [55]. The authors suggest the use of a single interaction approach (like in TRAX model [56]) to reproduce energy electron spectra from GNP.

A second observation is that the large majority of microdosimetry studies are performed by sending IR directly on the nano-object (hypothesis that 100% of the GNP are hit by the radiation). This approach can lead to overestimation of electron emission since not all IR will encounter a GNP in a realistic geometry. This is especially the case for charged particles since their higher LET means that the number of projectiles required to achieve a given dose of radiation is smaller than in the case of photons. Heuskin et al. [48] demonstrated that the interaction probability of GNPs with the incident proton beam is very low when considering a realistic cell model. Based on [30] which reported a significant 0.14 DEU, they calculated that only a 10^−6^–10^−5^ fraction of the total nanoparticle content interacts per Gy of radiation. Similarly, Lin et al. [57] reported very low interaction probabilities per Gy for spread-out Bragg peak protons (≈10^−8^ to 10^−4^ for the 2–50 nm GNP range). In addition to this energy-dependency, the probability scales with the third power of GNP radius as illustrated in [58] in the case of photon irradiation. Consequently, this highlights the need to consider the effect of Poisson statistics, describing the interaction probability of projectiles in a given volume. 

The last observation is the need for more realistic geometry models since theoretical works are mostly performed in conditions that do not reflect the in vitro experimental conditions. In fact, GNPs are known to be distributed heterogeneously throughout the cell volume under both passive and active targeting [15,30,59]. Moreover, it was reported that NPs typically aggregate and form clusters within cells [30,60], increasing the heterogeneity of the NP distribution. Therefore, the reported short ranges of LEE imply the energy deposition mostly in the vicinity of GNPs themselves, where the probability to find another GNP is high. This may result in a non-uniform spatial distribution of the dose enhancement within cells. In a recent article, it was demonstrated that NP clustering reduces the proportion of total energy absorbed in the surrounding water [61]. A 32% decrease in dose enhancement was observed with NP clusters evidencing that tightly clustered NPs are less effective at enhancing radiation damage than widely dispersed nanoparticles. Moreover, the short-range of LEEs reported in literature has to be put in perspective with the cellular GNP location. Many authors [42,44] concluded that GNPs have to be located inside the nucleus to produce a significant increase in cell death. However, only a few works have described a nuclear localization of ultrasmall GNPs while the vast majority of studies evidenced a localization in the cytosol (at distances of the order of micrometer from DNA) [26,30,62]. In addition, the emitted electrons have to pass through the coating layer that surrounds GNP before reaching water. Spaas et al. [63] have measured an average 5.2% loss of radiosensitization enhancement per nanometer of PEG coating after 200 kVp irradiation. Thereby, the coating effects have to be taken into account in simulation codes like in [64], especially for charged particle irradiation for which the effect will be more significant due to the main production of short-range electrons. Finally, the cell exposure to IR generates a succession of different responses that are not limited to physics, as discussed previously. Thereby, Monte Carlo simulations need to be extended to the subsequent chemical and biological stages. That is the case in TOPAS-nBio, an extension of Geant4, which includes track structure simulations in cell and sub-cellular geometries, as well as a large set of chemical reactions.

Although there is room to progress in our understanding of the physical mechanism, some evidence in literature is contradictory with a dose enhancement. It has been demonstrated that nanodiamonds can achieve a significant sensitization effect while no dose increase is theoretically predicted for carbon-based nanomaterials [65]. While a physical dose enhancement occurs under NPs irradiation, by no means it explains on its own the sensitization effect observed in vitro. Therefore, other mechanisms have to be involved.

### 3.2. Chemical Enhancement

As discussed here above, the energy deposition leads to the formation of various ionized and excited water molecules (ROS). Due to their great chemical instability, ROS are capable of interacting with all types of biological molecules leading to severe damages to cellular components. Compared to physical mechanisms, chemical enhancement has not been extensively investigated. It was reported that the presence of radical scavengers during the irradiation significantly reduces the enhancement effect in many different cell types [30,31,47]. Moreover, when radiosensitization experiments are conducted in anoxic conditions (pO_2_ < 1%), no significant enhancement is observed [18]. These studies strongly suggest a ROS contribution to the GNP radiosensitization. It is well known in literature that increasing cellular ROS level can lead to cell death via various mechanisms [66]. In addition to their well-known effects on DNA (base oxidation, single strand breaks, double strand breaks, etc.), ROS also affect other biomolecules such as lipids. Lipid peroxidation is one of the primary consequences of a cellular oxidative stress. This chain reaction consists of oxygen addition on unsaturated fatty acid (RH) to form organic peroxides (ROOH) after the initiation of the reaction by ROS. Therefore, plasma membrane phospholipids and organelle membranes such as the mitochondria can be oxidized, leading to biophysical changes that disturb membrane and organelle function. Oxidation of cardiolipin (a mitochondria-specific lipid) results in a weaker interaction with cytochrome C, an essential protein of the respiratory chain pathway. Data suggest that the increase in oxidized cardiolipin content triggers cytochrome C release from the mitochondrial intermembrane space into the cytosol leading to the apoptosome formation and to caspases activation, hence to apoptosis [67]. Moreover, lipid peroxidation leads to the production of additional reactive species such as 4-hydroxy-2-nonenal (HNE) which contributes to toxicity through protein cross-linking [68]. HNE was also suspected to act as an upstream modulator between oxidative stress and endoplasmic reticulum (ER) stress since a higher HNE levels can trigger unfolded protein response (UPR) [69]. ROS can also induce apoptotic signaling through the hydrolysis of sphingomyelin to ceramide, mimicking the effects of the cytokines TNF-α and IL1-β [68,70]. These examples demonstrate how an increase in ROS level can lead to cell death and therefore the importance to investigate their potential contribution to the radiosensitization effect. 

To identify the chemical enhancement contribution, it is important to define criteria enabling the distinction from the physical enhancement. According to that detailed in the previous section, physical enhancement has an energy dependence with a maximum dose enhancement in the range 10–200 kV while MV photons only give a negligible value (Figure 2B). In addition, theoretical predictions evidence a maximum of 1 DEU.WP^−1^ for kV photons meaning that no significant dose enhancement is expected with gold concentration below 0.01 WP. These characteristics will be used to ensure that the experimental results obtained in this section cannot be explained by a physical enhancement.

In literature, two main mechanisms by which nanoparticles may enhance ROS generation are described: The first one directly refers to physical enhancement since some LEEs have enough energy (>32 eV) to ionize oxygen-based molecules surrounding GNPs, leading to ROS formation.The second one is through catalytic processes. In contrast to the widely accepted theory that GNPs are chemically inert materials, increasing evidence showed that GNP surface is electronically active and capable of catalyzing chemical reactions [71]. Nano-objects have large surface/volume ratio and so a great amount of surface atoms that are not fully coordinated. Therefore, these surface atoms can interact with reagents and/or stabilize reaction intermediates leading to a decrease in reaction energy barriers. It was demonstrated that GNPs can interact with H_2_O_2_ or O_2_ to transform organic compounds through catalytic processes [38]. Liu et al. [72] evidenced hydroxyl bond formation between water and GNP leading to a decrease in water dissociation energy. Similarly, some studies claimed that GNPs catalyze the formation of ROS through a surface interaction with molecular oxygen, which facilitates surface-mediated transfer of electrons [73,74]. This may explain the oxidative stress reported in cells incubated with GNPs in the absence of IR [26,75,76,77]. In combination with IR, this catalytic property can be enhanced by interacting with the highly reactive environment generated by the irradiation. Moreover, electrons emitted from GNPs that have energy lower than usual water ionization energy, could lead to ROS formation driven by the decrease in ionization potential.

To study the chemical enhancement, the majority of studies focus on the production of ^●^OH radical since it is considered as the most powerful oxidant (E°(^●^OH/OH^−^) = 1.90 V at pH 7) of water derivatives [78]. The most practical way to quantify this ROS production is to use the hydroxylation reaction of coumarin 3-carboxylic acid (3-CCA) to form fluorescent 7-hydroxycoumarin 3-carboxylic acid (7-OHCCA) molecules. Significant higher enhancement values were observed using this reaction compared to predicted one if only the physical effect would take place (8 DEU.WP^−1^ for 100 kVp X-rays [79]; 60 DEU.WP^−1^ for 17.5 kVp X-rays [80]). Moreover, these studies evidenced the importance of NP surface. Cheng et al. [79] demonstrated that 3 nm GNPs are more efficient to produce ^●^OH than 30 nm highlighting a significant correlation (*p* < 0.01) between chemical enhancement and GNP surface. Furthermore, they observed no enhancement when the GNP surface was covered using an inert silica layer. Gilles et al. [80] reported that GNP functionalization using organic polymer markedly decreases the hydroxyl radical production under X-rays. This decrease in ROS production is proportional to the number of atoms in the coating, suggesting two explanations. First, the physical enhancement is expected to be responsible for electron emission from the GNPs at low X-ray energy. If these electrons are scavenged by the coating before they can react with water to produce ROS, hydroxyl radicals should be less abundant in solution. Secondly, they proposed that, at the nanoparticle–water interface, the coating layer partially prevents the decrease in water ionization energy. In addition, a dose rate dependency in the measured enhancements was evidenced [81], which disagreed with the physical mechanism since this parameter does not affect how photons are absorbed by GNPs or how charged particles lose their energy in GNPs. In the frame of a catalytic process, ^●^OH radical production is expected to occur in close vicinity of GNP surface leading to the formation of localized concentrated ^●^OH area. Since the radical density in this region is proportional to the dose rate, the probability of ^●^OH recombination reaction increases. This leads to the formation of H_2_O_2_ molecules, species not detected with 3-CCA explaining the reported decrease in chemical ^●^OH enhancement with the dose rate. 

In the same way, other groups evidenced a chemical enhancement by focusing on other types of ROS. Misawa et al. [82] studied the enhancement effect of five different GNP sizes in terms of superoxide radical production. Although they used solutions of 0.001–0.1 WP (concentrations too low to cause any physical enhancement), they reported up to a 7.86-fold increase in superoxide anion production upon irradiation with 100 kVp X-rays. The authors attributed the observed enhancement to secondary X-ray fluorescence and Auger electron emissions, which cause water radiolysis. However, evaluation of the relationship between ROS production and GNP size highlighted that smaller GNPs yielded higher levels of ROS compared to bigger ones suggesting a key role played by the NP surface (catalytic process). The authors studied the size effect by using the same gold WP in all conditions meaning that smaller GNPs, more numerous, expose a larger total surface than the bigger ones. 

Taken together, all the aforementioned studies evidenced that GNPs can enhance the radiation effects through an increase in ROS production. However, the small number of studies on this particular topic limits our complete understanding of this chemical mechanism. Future studies have to use methods enabling real-time, in situ measurement of ROS. This constitutes a challenge considering their short live time (ranging from nanoseconds for ^●^OH to milliseconds for H_2_O_2_). Although techniques like Raman spectroscopy have successfully reached this goal [83], the complexity to implement these analyses on clinical irradiation facilities remains the limiting factor. However, it has to be noted that this kind of experimental setup would be extremely interesting in order to investigate chemical processes in radiobiology context independently of GNP presence. This technique could be used to determine real-time particle range in protontherapy treatments, a current challenge in hadrontherapy development. In addition to radical production measurements, it will be interesting to develop methods enabling to probe surface atoms during catalysis processes. Ultrafast extended X-ray absorption fine structure was recently suggested to identify changes in oxidation/reduction state of surface atoms during irradiation [38].

Further chemical investigations have to be performed to fully understand the influence of GNPs in radiolysis processes. Although a catalytic ROS production can explain the higher dose enhancement reported in in vitro studies compared to the physical enhancement predictions, these radicals will be created in cancer cells, which possess a high redox status. In fact, it was reported that cancer cells exhibit higher reduced glutathione and antioxidant defense enzyme contents leading to resistance to oxidative stress [84]. Thereby, the GNP-induced radiosensitization effects have also to be studied from a biological point of view for a full understanding. 

### 3.3. Biological Enhancement

As explained earlier, the success of modern radiotherapy is influenced by five factors. If GNPs can affect this 5 R at the molecular and cellular levels, they could modify the cell response to radiation. Although a potential impact of NPs on biological pathways has been recognized, only a relatively small number of research groups have investigated it. Evidence obtained to date for a “biological enhancement” can be classified according to the 5 R.

#### 3.3.1. Repair

DNA damage-sensing proteins have been shown to concentrate at sites of DNA double-strand breaks (DSBs) after exposure to IR. This accumulation of specific proteins enables the formation of immunofluorescently labelable nuclear domains, known as radiation-induced foci (RIF). RIF measurements are routinely used to quantify DSB and evaluate repair kinetics after different treatments [85]. It is widely recognized that the number of unrepaired RIF is directly correlated to cell death [86,87]. Many groups have investigated the GNP impact on DNA damage generation and repair through RIF quantification. Several authors reported higher RIF numbers when cells were irradiated in the presence of GNPs [20,88,89,90,91,92]. However, some of these studies were performed with X-ray doses ranging from 2 to 6 Gy, which correspond to a range where RIF-dose relationship reached a saturation indicating multiple DSB coalescence into single RIF [85]. Unfortunately, the loss of relationship between DSB and RIF prevents a clear understanding of these results. Although other groups worked in the linear RIF-dose range, they investigated GNP impact on DNA repair by analyzing the RIF number at only one or two time-points post-IR [52,91,93]. This approach only offers a static view of a dynamic process lasting several hours. Therefore, it is still complex to conclude if the higher RIF number reported 24 h after irradiation in cells incubated with GNPs is due to DNA repair delay or to a higher level of persistent DSBs. In a recent article, we analyzed the DNA repair process in cell containing GNPs exposed to 1 Gy X-rays through RIF measurement at several time points ranging from 15 min to 24 h post-IR [26]. The kinetics of the repair process was analyzed using a mathematical formalism demonstrating that the presence of GNPs did not influence the total number of DSBs produced per cell but led to a 25% decrease in the repair process rate. 

Although some groups attributed the higher RIF level at 24 h post-IR to a higher DNA damage induction caused by physical and chemical enhancement, the reality seems to be more complex. In fact, Cui et al. [93] did not observe any significant difference in DNA damage at 30 min post-irradiation suggesting that GNPs did not increase the amount of DSB produced by the irradiation. Jain et al. [52] and we [26] reached the same conclusion by reporting no impact of 1.9 and 10 nm GNPs on the DSB production in MDA-MB-231 and A549 cells, respectively. 

An alternative to RIF quantification for the DNA repair investigation is the analysis of overall changes in related gene expression or protein activity. Li et al. [94] reported downregulation of BRCA1, Hus1, ATLD/HNGS1, and AT-V1/AT-V2 mRNA levels, which are DNA damage response genes, in MRC-5 lung fibroblasts exposed to 20 nm GNPs. Similarly, Schaeublin et al. [98] observed that exposition to 2 nm GNPs leads to significant downregulation of various genes involved in DNA repair including ATM, RAD21, and MRE11A. The changes in gene expression reported in the literature are summarized in Table 2. At the protein level, a decrease in the expression of DNA damage sensor proteins such as AKT [102] and Ku70 [103] (involved in non-homologous end joining pathway) or Rad51 [103] (involved in homologous recombination pathway) was reported in the presence of GNPs. Moreover, a significant reduction of thymidylate synthase (an enzyme involved in the production of deoxythymidine triphosphate, an essential precursor for DNA repair) abundance was also evidenced [104]. 

The reported delay in DNA repair coupled to the downregulation of DNA repair gene expression suggest that GNPs may interact directly or indirectly with regulators of genomic integrity. Since GNPs were usually not found in the nucleus but mainly located in cytoplasmic vesicles [103], additional works have to investigate this potential (in)direct interaction with key regulators.

#### 3.3.2. Redistribution 

The cell sensitivity to IR is influenced by their position in the cell cycle phases. It was reported that cells in the S phase are the most resistant while cells in late G_2_ and M phases are the most sensitive [7]. Some works did not evidence any significant change in cell cycle distribution when different cell lines were incubated with 1.4 [77], 1.9 [32,52], or 2.7 nm GNPs [93]. In contrast, various groups have evidenced the capacity of GNPs to induce cell-cycle arrest. The large majority reported a G_2_/M phase arrest coupled to a G_0_/G_1_ or S phase acceleration [25,28,95,96,105,106,107]. Changes in gene and protein expression reported in the literature enable a better understanding of this cell cycle redistribution. Roa et al. [105] observed an increase in cyclin B1 and E expression as well as a decreased expression of cyclin A, which is in agreement with the aforementioned G_2_/M phase arrest. *Cdh1* gene was significantly upregulated leading to the silencing of mitotic cyclin-dependent kinase 1 (CDK1) activity [95,96] while MAD2 (protein that regulates the mitotic spindle checkpoint), HsT17299 (encodes for cyclin B2), and *CCNB* (encodes for cyclin B1) gene expressions were downregulated (Table 2). These changes in gene expression can lead to G_2_/M phase arrest. In addition, downregulation of the expression of *MCM* gene family (MCM2, MCM5, and MCM6) was reported, resulting in limiting the transition from the G_0_ to G_1_/S phase [96,97]. In another study, Bhattacharya et al. [108] showed that citrate GNPs induced the overexpression of p21 and p27 proteins in various multiple myeloma cell lines leading to an arrest in G_1_ phase. Finally, a decrease in cells in G_2_/M phase was observed when GNPs coated with a nuclear localization sequence are incubated with different cell lines [109,110].

Despite increasing evidence that GNPs can induce cell cycle arrest, it is still not possible to draw conclusions about it due to the lack of data regarding molecular mechanism of action and the great variability in experimental conditions used in the studies. GNP size could play a role since no significant cell cycle interference was reported in the majority of works performed using very small GNPs, e.g., <3 nm [32,52,77,93].

#### 3.3.3. Repopulation 

Subpopulation of repopulating cells has been identified as the main cause of radiotherapy failure. GNP ability to slow down the cell proliferation goes hand in hand with the cell cycle arrest and has been demonstrated by different groups [26,32,76,108,111]. We reported a 1.4-fold increase in doubling time when A549 cells are incubated with 10 nm GNPs [26]. Studies have evidenced the importance of biology in this effect since Aurovist^TM^ (1.9 nm spherical GNP) exposure can lead to cell growth delay or to no significant change in proliferation rate depending on the cell line of interest [32,77].

The molecular basis of the anti-proliferative effect is complex due to the multiple molecular pathways involved [112]. GNPs have been shown to have a huge impact on growth factors (VEGF-165, VEGF-A, FGF, etc.) through activity inhibition and/or expression reduction [100,112,113,114]. This can lead to inhibition of Src kinase [115] and VEGF-A/VEGFR pathways and to subsequent inhibition of AKT and ERK phosphorylation [116]. Moreover, Satapathy et al. [114] demonstrated that GNPs can downregulate the levels of pro-angiogenic factors (Ang-1 and Ang-2) and decrease the production of inflammatory cytokines from fibroblasts. GNP can also affect actin cytoskeleton, which participates in cell proliferation regulation. It was hypothesized that integrin-mediated cell adhesion could play a role since downregulation of specific integrins, such as α_v_β_3_ [106], and genes involved (PIP5K2B, ACTN1 [97]) were reported. *ACTG1*, a gene encoding for actin γ was also found significantly downregulated (Table 2). Lower expression levels of matrix metalloproteinases 2 [114] and 9 [101], enzymes that degrade extracellular matrix components were evidenced, limiting the cancer cell spread. Finally, Wu et al. [101] observed a downregulation of ICAM-1 in cells containing 5 nm GNPs causing a strong inhibition of cell invasion potential. Altogether, these findings highlight the impact of GNPs on actin polymerization and assembly as well as on the interaction of cells with the extracellular matrix. The disturbance of these processes can lead to a decrease in cell proliferation rate in vitro and cancer cell metastasis in vivo.

#### 3.3.4. Reoxygenation

Despite its known key role in radiation therapy success, the importance of oxygen in GNP enhancement effect was poorly investigated. It was reported that radiosensitization effect is greater under normoxia than under hypoxia conditions (pO_2_ < 1%) [18,93]. Analogously, many groups have evidenced a reduced enhancement effect when cells containing GNPs are exposed to IR in the presence of radical scavengers [30,31,47,75]. These two sets of observations highlight the key role that oxygen seems to play in the radiosensitization effect.

It was demonstrated that GNPs can increase the intracellular ROS level in the absence of radiations. This was evidenced through flow cytometry analyses [26,75,76,77] as well as via the significant upregulation of oxidative stress related genes such as FOS or JUNB (Table 2). The increase in ROS level leads to the induction of a cellular oxidative stress, which can damage essential biological targets. Taggart et al. [75] reported the oxidation of mitochondria lipids resulting in a decrease in mitochondria membrane potential. We also demonstrated that this change in mitochondria membrane potential follows the time-dependency of ROS level fluctuations suggesting a link between these two events [26]. However, additional studies are needed to understand the exact link between oxidative stress and mitochondria potential disruption since the latter is associated to superoxide anions leaking into the cytosol, which can in turn induce an oxidative stress. In addition to their impact on mitochondria, several authors reported the induction of ER stress following GNP incubation [103,117,118]. Yasui et al. [103] observed that GNPs increased the expression of IRE-1α, BiP/GrP78, Calnexin, and Ero-1α as well as PERK phosphorylation, which are several ER stress-related molecules. Although no direct correlation with oxidative stress was demonstrated in the previous work, the link was hypothesized since increased ROS level induced by other metallic NPs has been extensively shown to induce ER stress [117].

While critical when they are produced in large amounts, low levels of ROS are used as intracellular messengers in various signaling pathways. Mammals have evolved a series of antioxidant defenses in order to control the ROS level and protect vital biomolecules from their deleterious effects. These antioxidant defenses go from small endogenous molecules (such as reduced glutathione) that can directly scavenge ROS to complex enzymes that can repair the modifications/damages caused by ROS. It was reported that cancer cells exhibit higher reduced glutathione and detoxification enzyme contents leading to a resistance to oxidative stress [84]. In the light of this, the GNP impact on the cellular antioxidant defenses have to be investigated to understand how GNP can induce an oxidative stress in cells that are equipped to deal with it. Our group demonstrated that GNPs have the ability to inhibit thioredoxin reductase (TrxR) and gluthatione reductase (GR), two main redox reactions regulators, in cancer and normal cells [26,119,120]. TrxR and GR catalyze the reduction of their oxidized substrate (thioredoxin and glutathione, respectively) enabling to regenerate the substrate to their reduced antioxidant form. Moreover, these enzymes regulate the activity of other key players in regulating the redox status such as protein disulfide isomerase [121], which is suspected of contributing to the GNP radiosensitization effect [75]. Thereby, GNPs decrease the available pool of cellular reduced antioxidant substrates by combining an oxidative stress with the inhibition of reductase enzymatic activity. Moreover, Negahdary et al. [122] reported a significant decrease in catalase and glutathione peroxidase activities following GNP incubation. These enzymes catalyze the conversion of hydrogen peroxide (a harmful by-product of many normal metabolic processes) into less-reactive gaseous oxygen and water molecules. The inhibition of these enzymes in turn increases the oxidative stress. 

Upregulation of antioxidant defense-associated genes was also evidenced (Table 2). An increased expression of *MT1X*, a gene coding for one metallothionein that belongs to the cysteine-rich and heavy metal-binding protein family, was observed when fibroblasts were incubated with 20-nm citrate GNPs [97]. The same nano-object was reported to upregulate *SOD3*, a gene that encodes a member of superoxide dismutase protein family.

Altogether, these results demonstrate the GNP capacity to induce an oxidative stress by producing ROS that can damage important biological targets and by inhibiting different key antioxidant systems. 

#### 3.3.5. Intrinsic Radiosensitivity

A central issue in radiobiology is the marked difference in IR response observed between cells originating from different organs or patients. Since the pioneering work of Steel in 1989 [123], an increasing interest is growing worldwide for this problem. Nowadays, the origin of this variability remains poorly understood but it is mainly attributed to acquired mutations. Since cancer cells result from the accumulation of genetic alterations, it is not surprising to observe gene expression changes in cancer cells that could modify their intrinsic radiosensitivity compared to normal tissue. The most famous example is the highly frequent p53 mutation reported in cancer cells. Since p53 is a key mediator in the apoptosis regulation, mutation on this gene enables cancer cells to escape programed cell death and become subsequently radioresistant. Therefore, the design of drugs able to modify the cell intrinsic radiosensitivity through inhibition of specific targets that enhance the treatment resistance of cancer cell is interesting in clinic. Many groups have demonstrated the ability of GNPs to promote apoptosis [76,99,107]. Kumar et al. [107] showed an increase in the activation of caspases 3, 8, and 9, in DU145 cells incubated with 8 nm GNPs. Similarly, Wahab et al. [99] evidenced an upregulation of CASP3 and CASP7 genes caused by 15 nm GNPs.

In a recent article, we evidenced that the overexpression of *TXNRD1* gene (coding for TrxR protein) in tumor is associated to treatment resistance leading to a poor patient survival prognosis [119]. Since GNPs have the ability to inhibit TrxR and that cancer cells have differential basal *TXNRD1* expression, GNPs could have a potential impact on intrinsic radiosensitivity. However, the low number of cell lines used in this study prevents any significant correlation analysis between GNP enhancement effect and intrinsic radiosensitivity (usually determined as the clonogenic survival fraction at 2 Gy). To cope with this limitation, enhancement studies have to be conducted on a large number of cell lines using the same experimental conditions. 

Further biological investigations have to be performed to fully understand the GNP influence on cells without IR exposition. With this in mind, the “radioenhancer” community could benefit by discussing more with the “nanotoxicology” community that studies the effects of various metallic nanoparticles on cell homeostasis. It is interesting to note that most of the reported articles in this review regarding biological enhancement come from nanotoxicological studies. Since enzymatic inhibition by GNPs were reported, structural analyses of thiol/selenol active sites can help to identify potential new targets inhibited by gold ions/GNPs. 

### 3.4. Transdisciplinary Enhancement

Taken together, the aforementioned studies highlight the multidisciplinary aspects of the GNP radiosensitization mechanism, which is a complex mixture of physical, chemical, and biological processes that enhances the IR effects. A schematic representation of these processes is illustrated in Figure 4. When they reach a cell, GNP can interact with components of the plasma membrane and extracellular matrix. This leads to cell uptake through various processes including endocytosis or passive diffusion depending on the physico-chemical characteristics of the nano-object [124]. The high chemical reactivity of GNP surface promotes a catalyzed formation of ROS through the interaction with molecular oxygen (chemical enhancement). At first, the higher ROS level is addressed by the antioxidant defense system through the conversion of reduced antioxidant to their oxidized form. However, the regeneration of these substrates on their reduced antioxidant forms will be limited by the GNP action. Following endocytosis, endosomes containing GNPs fuse with lysosomes. The decrease in pH inside the vesicle probably triggers a restricted in situ degradation of the GNPs leading to a gold ion release [125]. Au^+^ ions could rapidly react with thiol and selenol groups constituting the large majority of endogenous antioxidant substrates and active site of antioxidant defense enzymes. The Au-S/Se bond thus created inhibits key enzymes including TrxR and catalase preventing reduced antioxidant regeneration and ROS direct conversion in water molecules, respectively. When the increase in ROS level surpasses the basal antioxidant level, a cellular oxidative stress is induced leading to damages in essential biological targets including mitochondria. This causes mitochondrial membrane depolarization that interferes with the electron-transport chain leading to cytoplasmic release of O_2_^●-^ radicals and impairment in the ATP production. Furthermore, mitochondria membrane oxidation leads to the opening of permeability pores resulting in the release of pro-apoptotic molecules, such as cytochrome C. This activates caspase 9, which triggers activation of caspases 3 and 7 and induces apoptosis. In addition, the decrease in Trx-(SH)_2_, the reduced form thioredoxin protein, can activate apoptosis signal-regulating kinase (ASK-1), also resulting in apoptosis induction [121].

The decrease in ATP content after GNP incubation [26] could interfere with various biological pathways requiring a lot of energy. In eukaryotic cells, the DNA repair machinery is highly ATP-dependent since it is necessary for a cascade of events requiring phosphorylation of several proteins taking part in DNA damage recognition and pathway initiation [126]. Moreover, essential enzymes for DNA replication are also ATP-dependent. It is the case for chromatin remodelers, which regulate chromatin accessibility, ensuring an access to the nucleosome in order to carry out the replication or the repair. Cell cycle distribution is also sensitive to ATP content. By inhibiting mitochondrial production of ATP, Sweet et al. [127] showed a significant increase in the G_1_ cell population induced by a low reduction in the ATP level while higher decrease (up to 35% in ATP content) elicited a G_2_-M accumulation. This highlights the ATP content dependency of checkpoints that regulate progression in the cell cycle. Finally, the oxidative stress also triggers a transcriptional response involving the activation of Mitogen-Activated Protein (MAP) Kinases, especially JNK and p38 MAPK [128]. This leads to the regulation of various gene expression (Table 2) that can affect cell cycle distribution, DNA repair machinery, and cell proliferation as previously discussed.

Thereafter, cells will be exposed to IR. The energy deposition in water or in GNP (physical enhancement) leads to an extra ROS production that increases the oxidative stress in a cell with limited resources to counteract them due to the inhibition of antioxidant defense systems. The DNA damage created by IR in cells that do not contain GNPs will be repaired. However, in cells containing GNPs, the lower ATP content and the reduced DNA repair gene expression limit the cell capacity to correctly repair DNA damages. This can lead to an increase in chromosome aberrations [129] and to a decrease the DNA repair rate [26] compared to cells that do not contain GNPs. It is widely recognized that the number of unrepaired double strand-breaks is directly correlated to cell death [86,87]. In Table 1, the large majority of enhancement studies reports an increase in α/β ratio for irradiated cells incubated with GNPs compared to irradiated cells that do not contain GNPs. This is characteristic of early responding tissue meaning that cells containing GNPs are intrinsically less capable of sub-lethal damage repair [5]. It is interesting to note that this observation is in agreement with the suggested impairment in DNA repair machinery. 

This process highlights the key role played by the oxidative stress in the GNP radiosensitization effect. The reported decreased radiosensitization effect under hypoxia compared to normoxia could be partially explained by a reduced oxygen concentration (which initiates the oxidative stress through the chemical enhancement) and the downregulation of the expression of membrane proteins involved in the endocytosis process [130]. This leads to a lower GNP internalization [18] which is directly correlated to antioxidant system inhibition [119]. 

In summary, this global mechanism highlights that GNPs play a radiosensitizer role by weakening the cells before the irradiation in addition to the radioenhancer role largely described in the literature through a physical dose enhancement. This section evidences the importance of considering NP–cell interaction in future studies. To go further in our understanding of the GNP enhancement effect, there is a need for harmonization of the experimental conditions [131]. Despite the great number of studies investigating the radiosensitization effect of GNPs, we have to face one important fact: the lack of data regarding key parameters and the great variability in experimental conditions reported in literature prevent drawing solid conclusions. To go forward, we have to:Precisely characterize the GNPs used in the studies including at least size, coating agent, shape, and gold content in the cell/tumor. This information is crucial to evaluate physical and chemical enhancement contributions to the whole mechanism.Establish a set of cell lines to benchmark in vitro data since every cell line has different DNA repair capacity, antioxidant defense basal expression, and intrinsic radiosensitivity.Investigate the influence of the time between NP incubation and IR exposition, a key parameter too often overlooked in the literature. Since biological enhancement is related to disruption of various cell processes, which takes time to occur, the time-dependency of GNP radiosensitization has also to be studied.Develop new methodologies to enable cross-correlative study between observed biological dysfunctions and GNP uptake in cell sub-populations. In a recent article, Turnbull et al. [104] correlated biological markers imaged using laser scanning confocal microscopy with gold content quantified in-situ using synchrotron X-ray fluorescence microscopy. This new methodology makes it possible to get rid of the global cellular response usually obtained by analyzing the entire cell population and to better understand processes occurring in given cell sub-populations.

## 4. How Do Mechanistic Findings Influence the Design of In Vivo Experiments?

The complex global mechanism introduced in Section 3.4 suggests that a careful parameter selection can optimize GNP therapeutic effectiveness by maximizing physical, chemical, and/or biological enhancements. The present section discusses how the aforementioned recent advances in mechanism may influence the design of future in vivo experiments.

### 4.1. GNP Size

The size of the nanoparticle is a key factor regarding the successful application of GNPs in radiotherapy treatment. The influence of this parameter on biodistribution and tumor uptake was extensively discussed in other reviews [132,133]. However, the aforementioned complex mechanism suggests that a careful size selection can optimize GNP therapeutic effectiveness. When the size of GNPs increases, the probability that the ejected secondary electrons lose their energy within the GNP increases, reducing the potential physical enhancement effect [134]. However, the increase in size can on the contrary be beneficial due to a higher interaction cross-section. In the same way, bigger GNPs will maximize the biological enhancement since they exhibit larger surface area enabling to maximize their partial digestion in lysosome. Contrastingly, small size GNPs exhibit large surface to volume ratio resulting in a higher chemical enhancement, which is based on the GNP ability to catalyze the formation of ROS. These trends should be put into perspective in light of the number of GNP internalized by cells, which is also size-dependent. It has been reported that injection of large GNPs (50–100 nm) in mice leads to their capture by the reticuloendothelial system (RES) leading to an accumulation in liver, spleen, and lymph nodes instead of in the tumor [135,136]. Contrastingly, the circulatory half-life of small GNPs (10–30 nm) seems to be longer resulting in a higher gold content in the tumor [62]. However, it was reported that ultrasmall GNPs (<10 nm) are rapidly eliminated from the body through rapid kidney filtration and urine clearance [133]. Therefore, the optimal GNP size is an intermediated one (5–30 nm) that will balance the higher tumor accumulation of small GNPs with the higher surface of larger nano-objects to optimize total intracellular gold surface. Figure 5 summarizes the situation. 

### 4.2. GNP Coating

The coating agent is an essential element of the nano-object design since it fulfills multiple roles: aggregation prevention, cell uptake improvement, and opsonization process modulation [136,137,138]. In the light of this, the current strategy is to cover GNP surface with large amount of polyethylene glycol (PEG, a biocompatible neutral polymer) enabling a long circulation half-life, a high tumor accumulation, and a low cytotoxicity. While the grafting of increasing amounts of PEG molecules confers a series of advantages in terms of in vivo biodistribution, it also decreases the physical and chemical enhancement effectiveness. Low-energy electrons emitted following the interaction between IR and GNPs can be scavenged by a large organic layer at the GNP surface resulting in a decrease in electron yield as observed by different groups [63,64]. Moreover, chemical enhancement through catalytic ROS production can be reduced as a function of coating thickness as reported by Gilles et al. [80]. Therefore, the thickness of the coating must be minimized while allowing a colloidal stability and preventing opsonization processes.

### 4.3. Inorganic Nature of the Nano-Object

Although this review focuses on gold nanoparticles, discussion on the mechanisms responsible for the radiosensitization effect opens the door to the use of other types of nanomaterials. From the physical point of view, the ideal NP should maximize the photon absorption. By applying the aforementioned methodology (see Section 3.1) to all the elements of the periodic table, it can be observed that gold is not the only material able to significantly increase photon absorption compared to water (Figure 6). Although high-Z materials such as hafnium (Z = 72), platinum (Z = 78), gold (Z = 79), or bismuth (Z = 83) constitute the bulk of the materials studied in the field, a second interesting high photon absorption area appears for atomic numbers ranging from 47 to 50 (silver to tin). It is also worthy to note that lanthanides (Z = 60 to 70) seem to be interesting materials since they absorb photons in a wider range of photon energies. Amongst these lanthanides, gadolinium (Z = 64) appears to be a good candidate due to an absorption window ranging from 9 to 70 keV and its MRI contrast agent properties, making it a potential theranostic tool [139].

From the biological point of view, the ideal NP needs to have the ability to release ions that can bind and the inhibit active site of antioxidant defense enzymes (mainly composed of thiol and selenol groups). To predict the reactivity of metallic ions with these functional groups, the hard soft acid base (HSAB) theory can be used. This theory uses Lewis acid-bases and classifies them on the basis of their relatively “hard” or “soft” character, a concept associated to polarizability (the ease with which electron density can be displaced or delocalized to form new covalent bonds). Hard character will be associated to low polarizability elements while highly polarizable elements will be defined as soft. This chemical hardness as a quantitative definition enables to classify every chemical element as a hard or soft acid or base [140]. The theory states that soft acids react faster and form stronger bonds with soft bases, whereas hard acids react faster and form stronger bonds with hard bases. Thiol and selenol groups are Lewis bases which have a low hardness parameter, i.e., are soft bases in HSAB theory. Thereby, the ideal NPs have to release metallic ions which are soft acids in the HSAB theory. According to [140,141], the potential candidates are Cu^+^, Ag^+^, Au^+^, Tl^+^, Pd^2+^, Cd^2+^, Pt^2+^, Hg^2+^, Ti^2+^, Fe^2+^, Pb^2+^ and metal atoms at zero oxidation state. Amongst these elements, silver NP grabs the attention since literature reported an important ion release from these NPs in lysosomes [125] and the previous dose enhancement discussion suggested an important physical contribution. However, the toxicity of these inorganic materials will also need to be considered.

### 4.4. Clinical Indications

From the physical enhancement point of view, an optimized photon absorption is obtained when kV X-rays encounter NPs, as illustrated in Figure 2B. These radiations are clinically used to treat solid tumors in two cases: Intra-operative radiotherapy (IORT): IORT is a radiation mobile radiotherapy technique consisting of delivering a large single X-ray dose between 30 and 50 keV to the tumor and the tumor bed (tissues surrounding the tumor up to 10 mm depth), at the time of surgical resection [142]. This technique reduces the need for post-surgical radiotherapy as well as the risk of recurrence in breast cancer patients [143]. In this context, an intratumoral injection of GNPs could enable to reach a high gold content in the tumor at the irradiation time, maximizing the encounter probability with the beam and enhancing the local dose deposition. Furthermore, surrounding healthy tissues are expected to receive lower radiation doses since photon attenuation by the GNPs would confine most of the dose to the tumor in which they are located.Brachytherapy: permanent brachytherapy is a radiation technique that consists of the insertion of radiative millimeter-sized seeds into the tumor. These seeds contain radioactive elements, which emit low-energy photons that are attenuated by a few micrometers of biological tissue. This enables to constrain the radiation inside the tumor, sparing healthy tissues. The main radioactive elements used in brachytherapy seeds are ^125^I (59.4 days half-life, photon emission peak at 27.5 keV), ^103^Pd (16.9 days half-life, photon emission peak at 20.7 keV), and ^131^Cs (9.7 days half-life, photon emission peak at 30.4 keV) [144]. As in the case of IORT, the presence of GNPs near the radioactive seeds could significantly increase the local dose deposition in the tumor. Alternatively, this approach would enable a decrease in total radioactivity administrated to the patient, as it would require lower activity per seed or injection of fewer radioactive seeds to achieve the same biological effect.

For deep tumors, MV photons are more often used. In this case, the chemical and biological contributions must be optimized to compensate for the lack of physical enhancement. Therefore, pharmacokinetic studies must be undertaken to determine the ideal timing between the GNP injection and the irradiation. This should correspond to the timing enabling the highest inhibition of cellular antioxidant defenses by the GNPs. Moreover, a maximization of the inhibition of defense systems and the amount of ROS generated can be achieved by modifying the inorganic nature of the nano-object (see Section 4.3).

From the biological enhancement point of view, GNPs can be useful in the treatment of overexpressing TrxR tumors which were reported to be more aggressive and associated to treatment resistance leading to poor overall patient survival [119,145,146]. Although the *TXNRD1* gene (coding for TrxR) is expressed in all cancer types, higher expression was found in testicular (median at 32 FPKM) and lung (median at 27 FPKM) cancers compared to other ones (average median 13 FPKM), as illustrated in Appendix A. Testicular cancers are mainly treated by chemotherapy. However, radiation therapy is the main treatment for lung cancer (77% of the patients receive radiation during their treatment according to [13]). It was reported that median survival time of lung cancer patients decreases from 102 to 44 months when tumors display high *TXNRD1* expression [119]. Thereby, GNP injection prior the radiotherapy treatment could inhibit TrxR and thus may increase the efficiency of treatment as well as the median patient survival time. Due to a positive correlation between TrxR inhibition and intracellular gold content [119], a significant radiosensitization can only be achieved with huge gold loading in the tumor. This suggests the use of intratumoral administration route which could give rise to some technical issues for moving tumors such as lung ones. 

## 5. Towards a Clinical Use

### 5.1. Current Status

Since the pioneering proof-of-concept in 2004 [15], hundreds of scientific papers were published regarding the potential use of GNPs as radioenhancer/sensitizer. Despite this great interest, no gold-based nanomaterial has yet received FDA approval in 2020. To date, several clinical trials involving GNPs are ongoing [58]. Table 3 lists the three gold-based nanomaterials under investigation in a context of cancer therapy:

Aurimune: CYT-6091 consists of 27 nm GNPs coupled to tumor necrosis factor alpha (TNFα), a cytokine involved in immune cell regulation. Although TNFα activation triggers various antitumor events, its uses in clinics is limited by severe associated side effects including hypotension and septic shock-like syndrome [147]. Therefore, its conjugation to GNPs enables enhanced anti-tumoral effects with reduced side effects in mice [148]. A phase I clinical study (NCT 00356980) was launched in 2006 to evaluate the maximum tolerated dose of CYT-6091 and its potential side-effects in patients with advanced unspecified solid tumors. Results have shown that CYT-6091 has the ability to target tumors and may be injected systemically at doses of TNFα that were previously reported as toxic (3-fold increase in maximum tolerated dose) [149]. In addition, an early phase I study (NCT 00436410) was launched in 2007 to investigate the CYT-6091 distribution in patients with primary or metastatic cancer undergoing surgery. To date, no results were published.NU-0129: NU-0129 is a 13 nm GNP coated with spherical nucleic acids, able to cross the blood–brain barrier. Once within the tumor, the nucleic acid component enables the targeting of Bcl2L12 gene, an upregulated gene in most human glioblastomas that plays a role in resistance to apoptosis. In 2013, Jensen et al. [150] demonstrated that injection of NU-0129 reduced the expression of Bcl2L12 in glioblastoma leading to a decrease in tumor progression in xenografted mice, without adverse side effects. Subsequently, an early phase I has started in 2017 to assess the safety profile of NU-0129 in patients with recurrent glioblastoma multiform or gliosarcoma. The clinical trial is still active, and no results have been already reported.AuroShell: AuroShell is a 150 nm silica nanoparticle coated with a thin layer of gold. Following absorption of near infrared light (NIR) by silica core, relaxation of electrons from gold produces a heat release that can be used for thermal ablation of cancer. In a preclinical study, Stern et al. [151] demonstrated the proof-of-concept by exposing mice bearing PC-3 prostate cancer tumor to NIR. They observed a 35 °C increase in temperature when the combination of AuroShell and NIR laser is used compared to a 14 °C increase for control (NIR laser only). Following these interesting preclinical results, Nanospectra Biosciences Inc. launched three different clinical studies. Two pilot studies evaluated the nano-object benefit in the NIR treatment of patients with refractory and/or recurrent head and neck tumors (NCT 00848042) and with primary and/or metastatic lung tumors (NCT 01679470). The third one is an active clinical study (NCT 02680535) recruiting patients to assess the use of AuroShell in the focal ablation of neoplastic prostate tissue by NIR irradiation. Although the first of these studies began in 2006, results have not been published yet.

It is interesting to note that none of the aforementioned clinical studies were conducted with the aim of using GNPs as radiosensitizers. Aurimune and NU-0129 used gold-based materials in a drug delivery context while AuroShell uses it for cancer thermal destruction. This absence in clinical trials is surprising in view of the large amount of in vitro and in vivo data published in the literature. Although significant contributions were reported, these studies were carried out in academic laboratories with scientific discovery goals and not with the aim of a translation towards the clinic. In recent years, increasingly complex GNPs are being developed enabling to also take advantage of chemotherapy drugs [152] or target tumor cells [31,89,91] or specific cell sub-structures [153,154]. However, this approach reinforces the fact that each group is working with their own gold nano-object, which possesses a given size, coating, and shape. Therefore, it remains difficult to learn from each other’s data and draw solid conclusions. Our limited knowledge regarding the mechanism of action of this radiosensitization effect is the best example. Although the unravelling of this mechanism is a necessary step prior to clinical studies, it is still being debated in the scientific community thus requiring additional works to go further in the clinic [155,156].

### 5.2. Steps and Challenges for the Clinical Translation 

The translation of GNPs into clinics is an expensive and time-consuming process that is associated to two main challenges.

#### 5.2.1. Large-Scale Manufacture

Compared to conventional formulation technologies that usually contain free drug dispersed in a given medium, the GNP technology is more complex in terms of manufacture. Although GNP are routinely produced by well-known wet chemistry protocols in academic laboratories worldwide, their Good Manufacturing Practices (GMP) large-scale production is challenging [157]. Preclinical works have determined optimal parameters for GNP efficacy, including size distribution, shape, or surface charge that should be preserved during the scaling-up process. Therefore, the radiosensitizers have to be manufactured at the same high-quality level and batch-to-batch reproducibility to ensure these given specifications, requiring expensive characterization facilities such as transmission electron microscopes. The challenge is even greater when surface modifications are considered. The grafting of a biomolecule for the tumor targeting requires multiple additional steps in the production process and complicate product quality control and quality assurance evaluation (amount of biomolecule/GNP, target recognition, etc.) [158]. Finally, GNPs have to be stable during long-duration storage ensuring the product quality at the time of clinical administration. 

To bridge the gap between small-scale academic laboratories and large-scale production, strong collaboration between academia and pharmaceutical companies or contract manufacturing organizations have to be established as early as possible in the product development.

#### 5.2.2. Extensive Toxicity Studies

No candidate medicine can be tested in humans before its safety profile has been proven in animal models. These preliminary toxicological studies enable an in-depth understanding of various preclinical aspects including pharmacokinetics, biodistribution, and toxicity in targeted and non-targeted tissues. Furthermore, this step is also an opportunity to assess biocompatibility potential depending on dose, establishing safe limits for further clinical trials [159,160]. This step is therefore of paramount importance in the case of GNP since no gold-based material is currently used clinically. In vivo studies have demonstrated a gold accumulation in organs associated to RES (liver, spleen, and kidneys) after intravenous injection [62,135,161]. Moreover, poor clearance from the body was reported with more than 90% of the gold loading still present in the liver 6 months post-injection [162]. This raises the question of long-term GNP toxicity for which histological alterations and inflammatory cytokine secretion were already reported [163,164].

These toxicological studies are governed by specific rules and regulations of Good Laboratory Practice (GLP), a quality system ensuring the uniformity, consistency, reproducibility, and reliability of non-clinical safety tests [160]. However, current approaches developed for conventional drugs are insufficient to fully assess the toxicity of nanomaterials. New specific approaches are thus being developed based on high-throughput screening techniques and in silico alternative test strategies [165]. Carrying out these specific tests in GLP-environment and the gap between the academic world and the regulatory aspects are hurdles that can be overcome by subcontracting the GNP development to GLP-certified contract research organizations. This approach has the advantage that project leaders do not have to bear the significant costs associated of maintaining GLP facilities. Cook and Payne [58] estimated that the cost associated to GLP toxicological studies ranges from $100,000 for a single low dose test using simple GNP to $10,000,000 if the study involves multiple doses of a nano-object including specific biological coating. It is interesting to note that once a gold-based nanomaterial will be approved in clinic, some GLP toxicity studies can be bypassed, reducing the costs for subsequent GNP developments. 

## 6. Conclusions

Over the past decade, the GNP radiosensitization field has undergone a growing interest for radiation therapy improvement driven by their unique physico-chemical properties. Although they were initially classified as solely physical dose enhancers, recent evidence casts doubt on this, highlighting an additional radiosensitizer role through chemical and biological impacts on cells. By identifying the biological targets of GNPs, this study opens new avenues towards the use of personalized medicine in radiotherapy. Moreover, this review contributes to address the question of radiosensitization mechanism(s) that remains a mandatory step towards the optimized clinical use of nanomaterials. While significant progress has been made, we outlined the main challenges to overcome to ensure a successful clinical translation. In view of the interdisciplinary nature of the phenomenon, this clinical transition will require the strengthening of collaborations between multidisciplinary scientific teams and medical/pharmaceutical partners. This will enable to bridge the regulatory gap, one of the main obstacles to current developments of nanomedicine. 

## Figures and Tables

**Figure 1 cancers-12-02021-f001:**
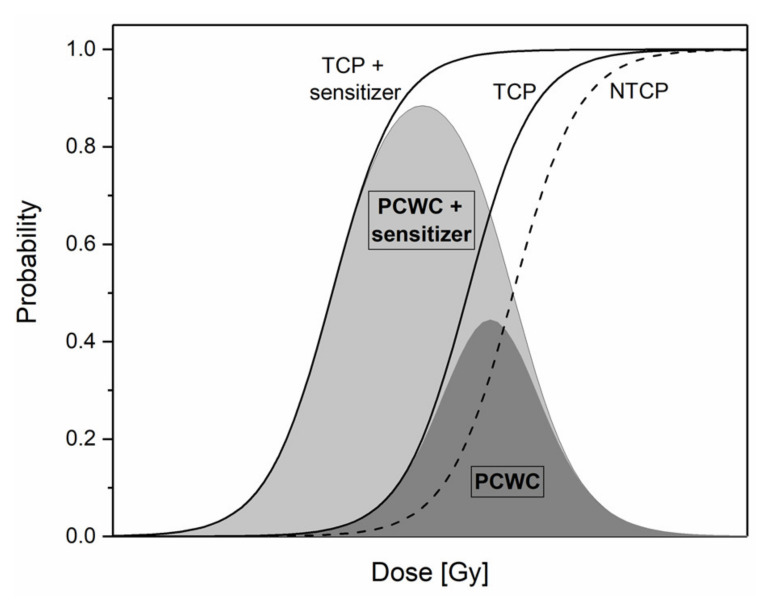
Schematic representation showing the action of tumor-targeted radiosensitizer on the probability of cure without complication (PCWC). The tumor control probability (TCP, solid line) and normal tissue complication probability (NTCP, dashed line) are shown as a function of the dose delivered to the patient. Two scenarios are illustrated with or without the use of a radiosensitizer and the associated PCWC is shown by the surface in each case. Adapted from [12].

**Figure 2 cancers-12-02021-f002:**
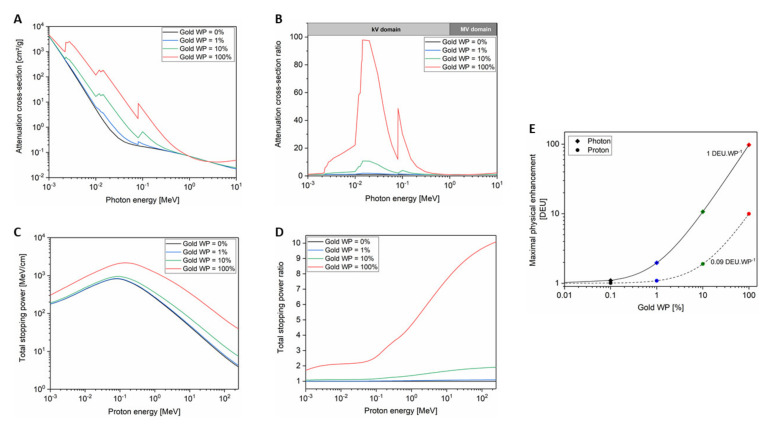
(**A**) Attenuation cross-section of photons in water and gold as a function of photon energy. (**B**) Ratio of gold attenuation cross-section on water attenuation cross-section depending on photon energy. (**C**) Total stopping power in water and gold as a function of proton energy. (**D**) Ratio of gold total stopping power on water stopping power depending on proton energy. (**E**) Maximal dose enhancement for photons (calculated for an energy of 15 keV) and protons (calculated for an energy of 200 MeV) depending on the gold weight percent. An enhancement of 1 DEU (dose enhancement unit) means that the delivered dose doubles when GNPs are present in the medium. Photon and proton data were derived from the NIST XCOM [36] and pstar [37] databases, respectively.

**Figure 3 cancers-12-02021-f003:**
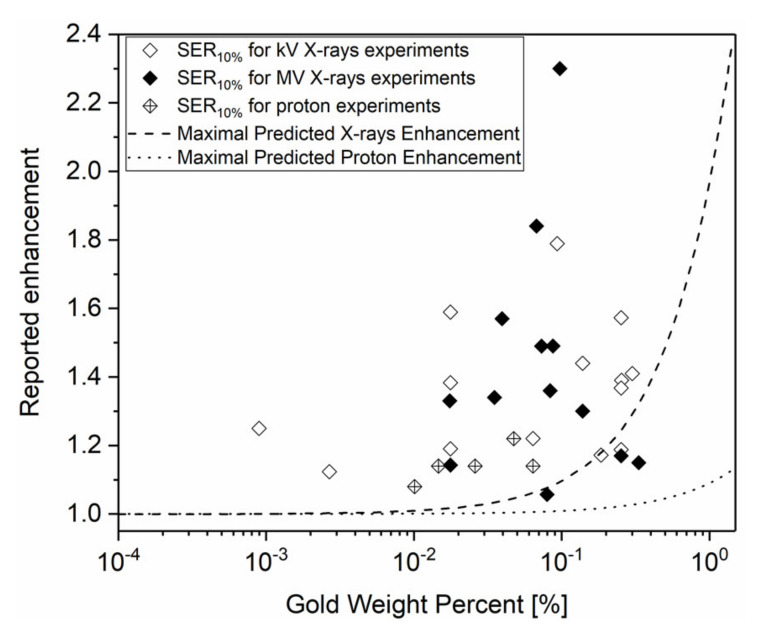
Comparison of observed in vitro experimental sensitization enhancement ratio (SER) with predicted dose enhancement values for GNPs studies. Dashed line represents maximal X-ray predicted enhancement calculated on a 1 DEU.WP^−1^ basis. Dot line represents maximal proton predicted enhancement calculated on a 0.09 DEU.WP^−1^ basis. Gold weight percent was calculated based on a theoretical eukaryotic cell volume of 3000 µm^3^ [54].

**Figure 4 cancers-12-02021-f004:**
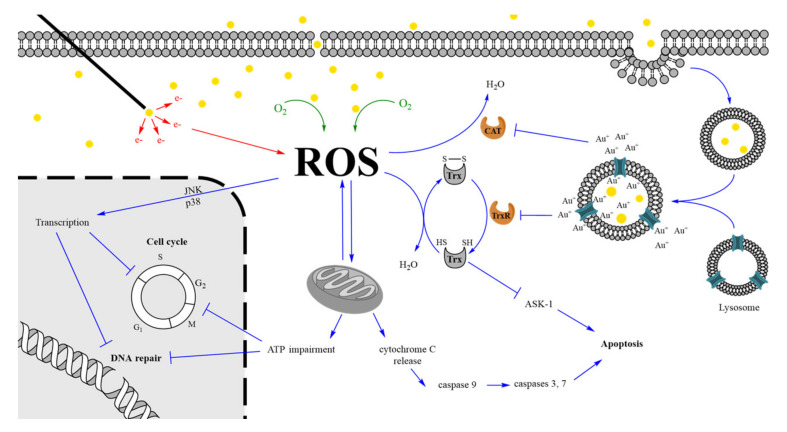
Schematic representation of transdisciplinary mechanism responsible for the radiosensitization effect of GNPs through interaction with cells. Physical dose enhancement, chemical enhancement, and biological mechanisms are illustrated by red, green, and blue lines, respectively. CAT = catalase; TrxR = thioredoxin reductase; Trx = thioredoxin; e- = electrons.

**Figure 5 cancers-12-02021-f005:**
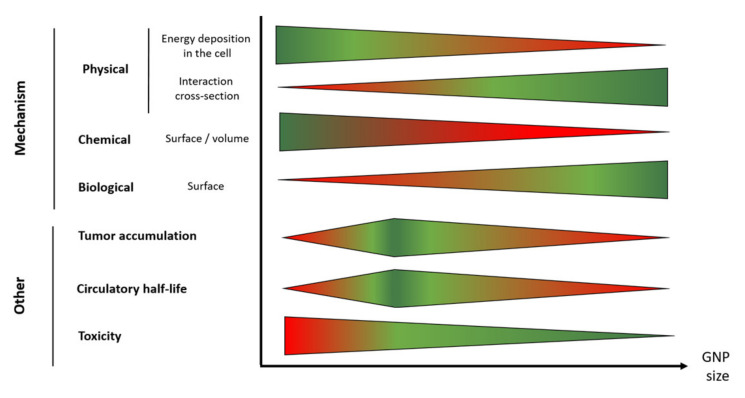
Schematic representation of the effect of GNP size on different factors involved in the radiosensitization effect. The triangle shows the evolution of the studied parameter as a function of the GNP size. The color refers to the action of this parameter on the radiosensitization effect (green = positive/will increase it; red = negative/will decrease it).

**Figure 6 cancers-12-02021-f006:**
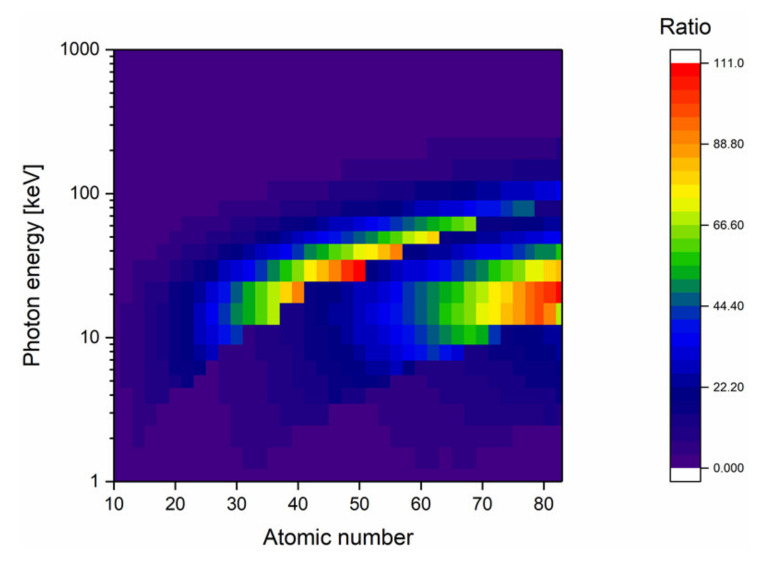
Comparison between photon absorption cross-section in water and in different materials. The ratio of absorption cross-section in a nanoparticle on absorption cross-section in water was plotted according to the incident photon energy and atomic number of the material of interest. Red areas indicate high dose deposition areas while purple ones correspond to no significant increase in photon absorption.

**Table 1 cancers-12-02021-t001:** Non-exhaustive list of in vitro studies of cell death enhancement by gold nanoparticles (GNPs). Only the studies that reported GNP cell uptake data and cell survival curves were selected. The results are alphabetically listed for the cancer type. The Sensitization Enhancement Ratio at 10% survival (SER_10%_) values were calculated from reported experimental data available in the corresponding publication/Appendix A. Fold change ratio in α/β is defined as the ratio of α/β from sample of interest (irradiated with GNPs) on α/β from control sample (irradiated without GNPs). An increase in fold change indicates a higher α/β when GNP are present during cell irradiation.

Cancer Cell Type	Cell Line	GNP Size (nm)	Coating Agent	Radiation Type	NP Uptake (#/Cell)	SER 10%	Fold Change in Ratio α/β	Reference
Brain	ALTS1C1	1.8	PEG - cRGD	117 kVp X-rays	9.00 × 10^6^	1.59	0.8	[16]
				250 kVp X-rays		1.38	1.6	
				662 keV γ-rays		1.19	0.8	
				6 MV X-rays		1.143	0.7	
				200 MeV protons Bragg peak		1.14	2.1	
	U87	1.9	Thiol	6 MV X-rays	3.45 × 10^7^	1.06	3.4	[17]
Breast	MCF-7	1.9	Thiol	6 MV X-rays	3.62 × 10^7^	1.36	6.3	[17]
	MDA-MB-231	1.9	Thiol	160 kVp X-rays	1.10 × 10^8^	1.39	28.8	[18]
				160 kVp X-rays	1.30 × 10^8^	1.41	2.7	
		16	Glucose	6 MV X-rays	5.30 × 10^4^	1.49	-	[19]
		49			9.40 × 10^4^	1.86	-	
Cervix	Hela	14	Citrate	220 kVp X-rays	2.90 × 10^3^	1.12	2.1	[20]
		50		105 kVp X-rays	6.00 × 10^3^	1.57	1.7	
				220 kVp X-rays		1.37	2.3	
				662 keV γ-rays		1.19	2.9	
				6 MVp X-rays		1.15	1.6	
		74		220 kVp X-rays	1.50 × 10^3^	1.17	2.0	
		113	-	6 MV X-rays	7.22 × 10^1^	1.34	-	[21]
		114	TAT peptide		1.95 × 10^2^	2.3	-	
		117	PEG		7.32 × 10^1^	1.57	-	
		128	Folic acid		9.56 × 10^1^	1.84	-	
Colorectal	CT26	4.7	PEG	6 MV X-rays	5.00 × 10^5^	1.33	-	[22]
	HT-29	24	Citrate	18 MV photons	7.13 × 10^4^	1.15	3.7	[23]
Liver	HepG2	16	Tirapazamine	50 kVp X-rays	6.50 × 10^2^	1.25	-	[24]
		150	-	160 kV X-rays	8.21 × 10^1^	1.79	-	[25]
Lung	A549	10	PEG	225 kV X-rays	1.90 × 10^5^	1.22	0.8	[26]
				1.3 MeV protons LET: 25 keV/µm		1.14	-	
		13	Glucose	6 MV X-rays	1.18 × 10^5^	1.49	-	[27]
Ovarian	SK-OV-3	14	Glucose	90 kVp X-rays	1.50 × 10^5^	1.44	-	[28]
				6 MV X-rays		1.3	-	
Prostate	DU-145	44	-	160 MeV proton Bragg peak	1.16 × 10^6^	1.15	-	[29]
Vulvar	A431	5	PEG	1.3 MeV protons	2.40 × 10^5^	1.08	-	[30]
		10			7.70 × 10^4^	1.14	-	
		10	EGFR antibody	1.3 MeV protons LET: 25 keV/µm	1.41 × 10^5^	1.22	-	[31]

PEG: polyethylene glycol; cRGD: cyclic arginine-glycine-aspartate motif; TAT peptide: GRKKRRQRRRPQ sequence; EGFR: epidermal growth factor receptor; GNP: gold nanoparticle; NP: nanoparticle; SER: sensitization enhancement ratio.

**Table 2 cancers-12-02021-t002:** Non-exhaustive list of gene expression changes in cells incubated with GNPs. Upregulated and downregulated genes are highlighted in green and red, respectively. When data are available in the publications, fold change is reported in the table.

Gene Category	Gene Symbol	GNP Size (nm)	Ligand	Fold Change	Reference
Cell cycle	CCNB	20	FBS	0.63	[94]
	CDH1	30	-	23	[95]
		45	BSA	2.22	[96]
	CDKN2B	1.4	TTPMS		[77]
	CDKN2C	1.4	TTPMS		[77]
	FBX04	1.4	TTPMS		[77]
	HsT17299	20	FBS	0.57	[94]
	MAD2	20	FBS	0.54	[94]
	MCM2	20	citrate		[97]
	MCM5	20	citrate		[97]
		45	BSA	0.59	[96]
	MCM6	20	citrate		[97]
	MEF2C	1.4	TTPMS		[77]
	RPA1	2	MES		[98]
	SESN1	2	TMAT		[98]
Cell death (apoptosis)	CASP3	15	citrate	1.4	[99]
	CASP7	15	citrate	1.5	[99]
	MCL1	18	citrate	1.56	[100]
Cell proliferation	ACTG1	20	citrate		[97]
	ACTN1	20	citrate		[97]
	ICAM-1	5	citrate	0.65	[101]
	MMP-9	5	citrate	0.37	[101]
	PIP5K2B	20	citrate		[97]
	TUBB2A	20	citrate		[97]
DNA damage signaling	APEX1	2	MES		[98]
	ATLD/HNGS1	20	FBS	0.92	[94]
	ATM	2	TMAT		[98]
	AT-V1/AT-V2	20	FBS	0.81	[94]
	BRCA1	20	FBS	0.65	[94]
	CHEK2	18	PAA	1.03	[100]
	Hus1	20	FBS	0.85	[94]
	MRE11A	2	TMAT		[98]
	MSH3	2	MES		[98]
	RAD9A	18	PAA	1.09	[100]
	RAD21	2	TMAT		[98]
	RAD51	2	MEEE		[98]
	RBBP8	20	FBS	0.75	[94]
	REV1	2	MES		[98]
Oxidative stress	FOSL1	1.4	TTPMS	8.2	[77]
	FTH1	18	citrate	1.52	[100]
	GSTM3	1.4	TTPMS	2.1	[77]
	HMOX1	1.4	TTPMS	23.7	[77]
	JUNB	20	citrate		[97]
	MT1X	20	citrate		[97]
	NF1X	20	citrate		[97]
	OSG1N1	1.4	TTPMS	10.2	[77]
	PRDX1	18	citrate	1.64	[100]
	SOD3	20	citrate		[97]

FBS = fetal bovine serum; BSA = bovine serum albumin, TTPMS = tri-phenylphosphine monosulfonate; MEEE = mercapto-ethoxy-ethoxy-ethanol; MES = mercapto-ethane-sulfonate; TMAT = tri-methyl-ammonium-ethane-thiol.

**Table 3 cancers-12-02021-t003:** List of clinical trials involving GNPs in oncology. NCT identifier is the clinical trial number referenced on the website “ClinicalTrials.gov”.

Name	Formulation	Company	Size (nm)	Admin. Route	Year	Status	Indications	NCT Identifier
Aurimune (CYT-6091)	TNF conjugated - PEGylated GNP	Cytimmune Sciences	27	I.V.	2006	Phase I, Completed	Unspecified solid tumors	00356980
					2007	Early phase I	10 specified cancer types	00436410
NU-0129	Spherical Nucleic Acid GNP	Northwestern University	13	I.V.	2017	Early phase I, active not recruiting	Gliosarcoma recurrent glioblastoma	03020017
AuroShell	Silica core coated with gold shell	Nanospecta Biosciences Inc.	150	I.V.	2009	Pilot study, completed	Head and neck tumor + NIR laser	00848042
					2012	Pilot study, terminated	Lung tumor + NIR laser	01679470
					2016	Phase I, active, not recruiting	Neoplasms of prostate	02680535

NCT: ClinicalTrials.gov identifier number; i.v.: intravenous; NIR: near-infrared irradiation.

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
