# Peer review of "Gold Nanoparticles as a Potent Radiosensitizer: A Transdisciplinary Approach from Physics to Patient"

_cancers, 2020, doi:10.3390/cancers12082021_

Round 1

Reviewer 1 Report

This article reviews the current state of knowledge by addressing how gold nanoparticles exert their radiosensitizing effects from perspectives of physical dose enhancement, chemical activation, to biological modulations. This review gracefully elaborates the discordance of radiosensitization of GNP between simulation and experimental data.

The impact of GNP on 5R discussion is one of the of the essences of this article, which has been rarely seen in previous reviews of the topics. Importantly, this article highlights the requirement of more toxicology studies and the collaborations between scientific teams and medical/pharmaceutical partners.  It certainly provides new insights in how we could go further toward clinical translation of GNP-mediated radiosensitization

There are some issues warranting further clarifications and discussions.

First, the authors narrow the scope on spherical GNP. However, the shape of GNP has been proved to significantly affect the cellular internalization, surface modifications, surface/volume ratio, and the related chemical activations.  Like the discussions in the section 3.1, the optimal selection of the size of GNP is the trade-off between chemical activation and cellular/tumor uptake. The shape of GNP should also be included for the discussion.

Second, the section 3.3 does not include the external-beam radiation therapy (RT), which is the main modality for clinical radiotherapy. Although brachytherapy/intra-operative RT using kV irradiation is more appropriate for physical dose enhancement, there has been several demonstrations of GNP-mediated radiosensitization with MV irradiation on the animal models. How those work, even with heterogeneous setting, achieved the successful radiosensitization may provide some insights for further translational work. 

Third, the discussions on size and coating in the section of 3.1 and 3.2 are all about systemic administration of GNPs. In contrast, the section 3.3 only talks about kV with intra-tumoral injection. This review would be much more comprehensive, if there are additional paragraphs for the literatures with MV and surface-modification of GNP for better pre-clinical application of GNP-induced radiosensitization.

Fourth, I agree with the authors that the use of nanomaterials other than gold deserves more attentions. Figures 6 is also a piece of great demonstration on how other metals interact incident photon. However, the section 3.4 right behind the clinical utilization is somehow distracting and beyond the focus of discussion for translational applications.

Finally, there are some abbreviations needing to be clearly explained when they are first used. For example, WP for weight percentage, DEU in the figure 2, etc. The cfr. in the line 220 is probably not frequently used.

Author Response

Please find enclosed the revised manuscript in which we took into account all the reviewer’s concerns. The modifications performed are highlighted in the main text. Hereunder, we answer point by point the reviewer’s questions and comments:

Reviewer 1:

  1. The authors narrow the scope on spherical GNP. However, the shape of GNP has been proved to significantly affect the cellular internalization, surface modifications, surface/volume ratio, and the related chemical activations. Like the discussions in the section 3.1, the optimal selection of the size of GNP is the trade-off between chemical activation and cellular/tumor uptake. The shape of GNP should also be included for the discussion.

We are grateful to the reviewer for this interesting comment. We have decided to restrict this review to spherical nano-objects (as indicated in the introduction, line 132) due to the lack of robust information available on the other shapes.

This review focuses on the mechanism of the radiosensitization effect. However, the most comprehensive study currently available on the shape effect does not show any impact of this parameter on the mechanism [1]. In this recent paper, the authors have studied the impact of spherical nanoparticles, nanospikes and nanorods of the same size and coating agent on KB cells irradiated by X-rays. They observed the induction of an oxidative stress and a cell cycle arrest in G2/M for the three types of nano-objects (as reported for spherical GNPs, see section 3.3 “biological enhancement”). However, the amplitude of these biological dysfunctions did not significantly vary with the shape of nanomaterials but seemed to be related to the level of nano-object internalization by cells that varies with the shape. This change in cell uptake would be responsible for the reported difference in SER since the authors no longer note any difference between conditions when SER is normalized by the quantity of gold internalized [1].

Thus, the shape of the nano-object seems to simply have an impact on the internalization capacity (as already discussed in many reviews [2-5]) without affecting the mechanism by which the radiosensitizing effect takes place. We therefore believe that, for the clarity of the manuscript, a discussion on the shape of the nano-object will not add value to the mechanistic message that this review is trying to convey.

  1. The section 3.3 does not include the external-beam radiation therapy (RT), which is the main modality for clinical radiotherapy. Although brachytherapy/intra-operative RT using kV irradiation is more appropriate for physical dose enhancement, there has been several demonstrations of GNP-mediated radiosensitization with MV irradiation on the animal models. How those work, even with heterogeneous setting, achieved the successful radiosensitization may provide some insights for further translational work.

We think the reviewer is referring to section 4.3 ("Clinical indications") and not to section 3.3 ("Biological enhancement") in his/her comment.

We believe that this comment stems from a misunderstanding of the purpose of section 4. Indeed, the purpose of this section is to discuss how recent advances on radiosensitization mechanism (section 3) may influence the design of future in vivo experiments. Therefore, section 4.3 discusses the preferential use of kV photons (brachytherapy and intra-operative radiotherapy) to maximize the physical effect and thus the radiosensitizer performance. Although many successful in vivo radiosensitization studies have been performed using MV photons, this type of ionizing radiation is not an optimized approach. For this reason, we do not discuss megavoltage external-beam radiation therapy in this section but it is discussed in detail in the rest of the manuscript.

To take into account the reviewer's comment, we have added a paragraph at the beginning of section 4 to clarify the role of this section and thus avoid misconceptions by the reader.

  1. The discussions on size and coating in the section of 3.1 and 3.2 are all about systemic administration of GNPs. In contrast, the section 3.3 only talks about kV with intra-tumoral injection. This review would be much more comprehensive, if there are additional paragraphs for the literatures with MV and surface-modification of GNP for better pre-clinical application of GNP-induced radiosensitization.

We think the reviewer is referring to sections 4.1 ("size"), 4.2 ("coating") and 4.3 ("clinical indications") and not to sections 3.1 ("physical enhancement"), 3.2 ("chemical enhancement") and 3.3 ("biological enhancement").

As in comment number 2, we believe that this comes from a misunderstanding of the purpose of section 4, i.e. to discuss how recent advances in mechanism influence the design of future in vivo experiments. Thus, a paragraph has been added at the beginning of the section to clarify its purpose and gain a better understanding.

The reviewer's suggestion to add a discussion on surface modification of GNPs does not seem to be appropriate in this section 4. The grafting of biomolecules improving tumor targeting will increase the radiosensitizing effect by maximizing the amount of internalized nano-objects and not the physical, chemical and biological effects responsible for the mechanism. For this reason, we propose in section 4.3 to use an intratumoral injection that enables the same expected effect (maximizing tumor gold loading) while avoiding the biodistribution issues that will always exist with a systemic administration of targeted nano-object. From a mechanistic and translation to the clinic point of view (focus of this review), surface modification is not the best strategy to adopt as discussed in several other parts of the manuscript:

* Section 3.1 (Physical enhancement): The emitted electrons have to pass through the coating layer that surrounds GNP before reaching water. Spaas et al. [6] have measured an average 5.2% loss of radiosensitization enhancement per nanometer of coating (PEG, organic moieties) after 200 kVp irradiation.

* Section 3.2 (Chemical enhancement): Gilles et al. [7] reported that GNP functionalization using organic moieties markedly decreases the hydroxyl radical production under X-rays. This decrease in ROS production is proportional to the number of atoms in the coating.

* Section 4.2 (GNP coating): In conclusion, the thickness of the coating must be minimized while allowing a colloidal stability to maximize chemical and physical enhancements.

* Section 5.2 (Steps and challenge for the clinical translation): The challenge is even greater when surface modifications is considered. The grafting of a biomolecule for the tumor targeting requires multiple additional steps in the production process and complicate product quality control and quality assurance evaluation (amount of biomolecule/GNP, target recognition, etc.) [8]. 

We believe that the discussion on surface modification that the reviewer suggests would not be appropriate to be added in section 4 because it would contradict the main message of this part of the manuscript. Nevertheless, it is addressed in the other sections.

  1. I agree with the authors that the use of nanomaterials other than gold deserves more attentions. Figures 6 is also a piece of great demonstration on how other metals interact incident photon. However, the section 3.4 right behind the clinical utilization is somehow distracting and beyond the focus of discussion for translational applications.

We thank the reviewer for this comment. Since the reviewer is referring to the figure 6, we think he/she is talking about section 4.4 (Inorganic nature of the nano-object) and not section 3.4 (Transdisciplinary enhancement). Therefore, we have swapped sections 4.3 and 4.4. This enables to end section 4 with a discussion on clinical indications of GNPs, thus linking to the next section "translation to clinics". We hope that the reviewer will find this new version more easily understandable compared to the original manuscript.

  1. There are some abbreviations needing to be clearly explained when they are first used. For example, WP for weight percentage, DEU in the figure 2, etc. The cfr. in the line 220 is probably not frequently used.

As suggested by the reviewer, we have provided an explanation for some concepts when they first appear in the text. For example, a sentence has been added to introduce the concept of DEU (line 212). In addition, the word “see” has replaced the words “cfr.” in the new version of the manuscript (lines 222 and 807).

Reviewer 2 Report

General evaluation:

The manuscript summarizes the current knowledge about the radiosensitizing potential of gold nanoparticles and their potential value for therapeutic approaches. First, the effects of irradiation on cells and radiotherapy to cure tumor patients are described, followed by chapters summarizing the radiosensitizing effects of gold nanoparticles on different tumor entities. Hereby, physical, chemical and biological effects are discussed and finally current challenges for clinical translation are outlined.

In summary, the manuscript describes an interesting and highly relevant topic aiming to review current knowledge about the potential of gold nanoparticles to act as radiosensitizers for the treatment of tumor patients. Unfortunately, throughout the manuscript references to the Figures and Tables are lacking which needs to be addressed as well as some additional points as described in detail below. Nevertheless, the review is well written and describes the topic in a comprehensible way, supported by very helpful and nicely presented Figures and Tables.

Specific evaluation:

Major points of criticism:

  1. Introduction, p. 3, equation (1) and Figure 1: The equation and Figure 1 are (almost) identical to Figure 15.3 and equation 15.5 in the book “Gold Nanoparticles For Physics, Chemistry And Biology (Second Edition)” by Catherine Louis, Olivier Pluchery, Chapter 15. Would it be possible to include appropriate references for the figure and equation?

Minor points of criticism:

  1. Multiple times (exactly 30 times) throughout the manuscript references are not appropriately provided (example: Background section, p. 3, ln 102-103 and 111: “Error! Reference source not found.” Appropriate reference Figure 1?) making it difficult to evaluate the manuscript. Please include correct references.
  2. Table 1: Please introduce the term “SER” in the Table legend.
  3. Background section, p. 3, ln 104: Please check the equation ????=??? .(1−????). What does the point behind “TCP” mean?
  4. Table 1: Please introduce the term “SER” in the Table legend.
  5. Please introduce abbreviations (e.g. PEG, TAT, FPKM) at first appearance in the text.
  6. Chapter 2.1, p. 9, ln 242-244: Please correct the sentence “The authors demonstrated that the electron emission from GNPs is 2.3 more important than in the case of gold surface and display a prominent peak below 100 eV.” Do the authors mean “… 2.3-fold more …”?

Author Response

Please find enclosed the revised manuscript in which we took into account all the reviewer’s concerns. The modifications performed are highlighted in the main text. Hereunder, we answer point by point the reviewer’s questions and comments:

Reviewer 2:

  1. Introduction, p. 3, equation (1) and Figure 1: The equation and Figure 1 are (almost) identical to Figure 15.3 and equation 15.5 in the book “Gold Nanoparticles For Physics, Chemistry And Biology (Second Edition)” by Catherine Louis, Olivier Pluchery, Chapter 15. Would it be possible to include appropriate references for the figure and equation?

We added the suggested reference to the figure legend as requested by the reviewer.

  1. Multiple times (exactly 30 times) throughout the manuscript references are not appropriately provided (example: Background section, p. 3, ln 102-103 and 111: “Error! Reference source not found.” Appropriate reference Figure 1?) making it difficult to evaluate the manuscript. Please include correct references.

We apologize for the inconvenience. The problem seems to come from the pdf version generated during the submission on the mdpi website. The problem has been addressed.

  1. Table 1: Please introduce the term “SER” in the Table legend.

As requested, the term SER has been introduced to the legend of Table 1.

  1. Background section, p. 3, ln 104: Please check the equation ????=??P .(1−????). What does the point behind “TCP” mean?

We thank the reviewer for noticing this error. The point does not represent anything and has been removed (line 106)

  1. Please introduce abbreviations (e.g. PEG, TAT, FPKM) at first appearance in the text.

We introduced the abbreviations in the new version of the manuscript as requested by the reviewer.

  1. Chapter 2.1, p. 9, ln 242-244: Please correct the sentence “The authors demonstrated that the electron emission from GNPs is 2.3 more important than in the case of gold surface and display a prominent peak below 100 eV.” Do the authors mean “… 2.3-fold more …”?

We thank the reviewer for this suggestion. The sentence has been modified in the new version of the manuscript.

Round 2

Reviewer 1 Report

I agree with the authors that the very first clinical trial of GNP-induced radiosensitization should start with KV irradiation and intra-tumor injection. Such the settings come along with more confidence from the mechanistic findings.

It has been well recognized that KV irradiation is more appropriate than MV photon/proton RT for GNP-induced radiosensitization from the view of physical enhancement. However, the characteristics of KV irradiation hinder its clinical indications.

It demands the revised addition that the authors do not provide additional discussions how chemical/biological mechanism might be utilized to compensate the lack of photon-GNP interactions if MV irradiation is planned to be used for broader clinical applications. 

Author Response

It demands the revised addition that the authors do not provide additional discussions how chemical/biological mechanism might be utilized to compensate the lack of photon-GNP interactions if MV irradiation is planned to be used for broader clinical applications.

We are grateful to the reviewer for this interesting comment. We have added a discussion on how chemical/biological mechanism can be optimized in order to compensate the lack of physical enhancement during MV irradiation (see lines 865 – 872).